# Camel nanobody-based B7-H3 CAR-T cells show high efficacy against large solid tumours

Dan Li[1], Ruixue Wang[1], Tianyuzhou Liang [1], Hua Ren[1], Chaelee Park [1], Chin-Hsien Tai [1], Weiming Ni[2], Jing Zhou [2], Sean Mackay[2], Elijah Edmondson [3], Javed Khan [4], Brad St Croix [5] & Mitchell Ho [1] ✉

Rational design of chimeric antigen receptor T (CAR-T) cells based on the recognition of antigenic epitopes capable of evoking the most potent CAR activation is an important objective in optimizing immune therapy. In solid tumors, the B7-H3 transmembrane protein is an emerging target that harbours two distinct epitope motifs, IgC and IgV, in its ectodomain. Here, we generate dromedary camel nanobodies targeting B7-H3 and demonstrate that CAR-T cells, based on the nanobodies recognizing the IgC but not IgV domain, had potent antitumour activity against large tumors in female mice. These CAR-T cells are characterized by highly activated T cell signaling and significant tumor infiltration. Single-cell transcriptome RNA sequencing coupled with functional T-cell proteomics analysis uncovers the top-upregulated genes that might be critical for the persistence of polyfunctional CAR-T cells in mice. Our results highlight the importance of the specific target antigen epitope in governing optimal CAR-T activity and provide a nanobody-based B7-H3 CAR-T product for use in solid tumor therapy.

To date, the US Food and Drug Administration (FDA) has approved six chimeric antigen receptor T (CAR-T) cells products against hematological cancers[1,2]. However, the successful application of these emerging cell-based therapies in solid tumors remains limited. B7 homolog 3 protein (B7-H3; CD276) is an attractive tumor antigen for immunotherapy due to its restricted distribution in normal tissues and high overexpression in multiple solid tumor types[3–7]. In humans, two isoforms of B7-H3 have been reported, 2IgB7-H3 (2Ig) and 4IgB7-H3 (4Ig)[8]. The 2Ig isoform has single extracellular V-like and C-like (VC) Ig domains, while the 4Ig isoform consists of two sets of VC-like Ig domains[9]. Knowledge about the function(s) of these two isoforms of B7-H3 in cancer remains limited[9,10].

More than 20 clinical trials including CAR-T therapy[11] targeting B7-H3 are in progress. B7-H3 CARs engineered with single-chain

variable fragments (scFvs) demonstrated encouraging antitumor activity and/or acceptable safety in preclinical studies against hematologic and multiple solid tumor types[3,12–15]. The first-in-human testing of scFv-based B7-H3 CAR-T cells has shown antitumor response against anaplastic meningioma and childhood diffuse intrinsic pontine glioma without severe side effects[16,17]. However, as yet no B7-H3 targeted therapies have garnered clinical approval.

The antigen-recognition domain is one of the most critical components in CAR-T cell therapy[18]. B7-H3 has two distinct epitope domains - IgC and IgV. While the epitope of most B7-H3 antibodies has not been described, one of the clinically tested antitumor antibodies (8H9) was found to bind to the IgV domain[19]. We and others recently demonstrated that the location where CARs bind in the ectodomain of a tumor antigen is critical for their activities[20–22]. In this regard,

[1]Laboratory of Molecular Biology, Center for Cancer Research, National Cancer Institute, Bethesda, MD 20892, USA. [2]IsoPlexis Corporation, Branford, CT 06405, USA. [3]Molecular Histopathology Laboratory, Frederick National Laboratory for Cancer Research, Frederick, MD 21702, USA. [4]Genetics Branch, Center for Cancer Research, National Cancer Institute, Bethesda, MD 20892, USA. [5]Mouse Cancer Genetics Program, Center for Cancer Research, National Cancer Institute, Frederick, MD 21702, USA. ✉e-mail: homi@mail.nih.gov

generating antibodies targeting the IgC and IgV domains of B7-H3 is helpful for identifying the potent epitope by which CARs are activated on B7-H3. Several rationales exist for using nanobodies in CAR-T cells, including their unique and potentially buried binding sites, small size, and ease of expression and production[23–25]. To obtain nanobodies that recognize diverse epitopes and have clinically relevant affinity without a need for further affinity maturation, we built large $V_H H$ nanobody libraries with great diversity (> $10^{12}$ pfu/ml total) from eight dromedary camels with four females and four males, with ages ranging from 3 mo to 20 y. We decided to systematically identify camel $V_H H$ nanobodies targeting the IgV or IgC domains of 4IgB7-H3 by phage display and produced nanobody-based CAR-T cells for preclinical testing.

In the present study, we demonstrate that B7-H3 CAR-T cells targeting the IgC domain are more active than those targeting the IgV domain in pancreatic ductal adenocarcinoma (PDAC) and neuroblastoma (NB) mouse preclinical models. The most potent nanobody-based CAR-T cells show inhibition of large tumor xenografts in mice with rigorous T-cell signaling and significant T-cell infiltration into the tumor. We also use single-cell functional proteomics and transcriptomics RNA sequencing to examine molecular determinants associated with the polyfunctionality of persistent CAR-T cells in mice. The epitope where an antibody binds on a tumor target is important for CAR-T activity. However, it is generally unknown what epitope works best for a tumor target. In this study, using large dromedary camel $V_H H$ phage libraries to isolate nanobodies that recognize all the major domains on B7-H3, we identify a previously undescribed epitope that enables the development of CAR-T cells with unusual activity against large tumors. The $V_H H$ nanobody-based B7-H3 CAR-T product described here is ready for clinical development in solid tumor therapy.

## Results

### 4IgB7-H3 is the dominant isoform in solid tumors

To characterize the B7-H3 expression pattern, we examined B7-H3 transcripts and found that B7-H3 was aberrantly expressed across various tumor types compared with paired normal tissues but restricted in vital organs (brain, heart, lungs, liver, and kidneys) (Fig. 1a). Moreover, the cell surface B7-H3 protein was highly expressed in multiple human tumor cell lines, belong to pancreatic adenocarcinoma (PADC), neuroblastoma (NB), ovarian serous cystadenocarcinoma (OV), lung adenocarcinoma (LUAD), epidermoid carcinoma, and hepatocellular carcinoma (HCC) (Fig. 1b). Patients with high B7-H3 levels exhibited poor overall survival compared to those with low B7-H3 expression in both PDAC and NB (Fig. 1c), which prompted us to test B7-H3 CAR-T cells in these two solid tumor types.

We further investigated the usage of two B7-H3 isoforms and found that, while acute myeloid leukemia (AML) lacked B7-H3 expression, 4Ig was the predominant form expressed in all tumor types analyzed except for ovarian (OV) cancer where 2Ig and 4Ig shared similar expression (Fig. 1d). To validate this, we designed a pair of B7-H3 specific primers and demonstrated that both 4Ig and 2Ig transcripts exist in B7-H3 positive (B7-H3⁺) tumor cell lines (Panc-1, IMR5, IMR32, 293T, and HepG2) rather than B7-H3 negative (B7-H3⁻) lymphoma cell lines (Raji, Jurkat, and Daudi) (Fig. 1e). However, using western blot, we observed only a 4Ig band in B7-H3⁺ tumor cell lysates with or without N-Glycosidase F Protein (PNGase F) digestion (~110 kDa or ~60 kDa) (Fig. 1f, panel i and ii), suggesting that the 4Ig is highly glycosylated. Using another anti-B7-H3 monoclonal antibody (mAb) which was previously reported to be able to predict both 4Ig and 2Ig protein in tissues[26], the 4Ig band was observed in B7-H3⁺ cell lysates (Supplementary Fig. 1). However, the "2Ig" band (~57 kDa) was also found in B7-H3⁻ cell lysates, indicating that the predicted "2Ig" was unlikely B7-H3. The soluble B7-H3 (sB7-H3) was previously reported as a biomarker for diagnosing patients with multiple solid tumors[27]. We demonstrated a significantly increased level of sB7-H3 protein in the

cell culture systems of Panc-1, IMR5, and IMR32 over time compared with culture media (CM) and IMR32 B7-H3 knockout (KO) culture supernatant (Fig. 1g). Taken together, these findings illustrate that 4IgB7-H3 is the dominant isoform expressed in multiple solid tumor types including PDAC and NB and can be developed as a pan-cancer target for cancer immunotherapy. However, the 2Ig isoform is rare, and further studies are necessary to validate the existence of the 2Ig protein in normal human tissues and other tumor specimens.

### Isolation of dromedary camel $V_H H$ nanobodies targeting B7-H3

To validate the 4IgB7-H3 isoform as a therapeutic target in cancer, we screened nanobodies against 4Ig from our eight large dromedary camel $V_H H$ nanobody libraries using phage display technology (Fig. 2a, see Methods). We obtained ten unique clones after three rounds of panning (Fig. 2b). All ten clones bound to human 4Ig protein, and half of them (referred to as C4, B12, G8, H5, and B2) showed cross-species binding to B7-H3 of monkey, murine, and rat but not to an irrelevant antigen (BSA) (Fig. 2b). Moreover, only three out of five purified monomeric camel $V_H Hs$ (C4, B12, and G8) bound to the native B7-H3 membrane protein expressed on tumor cells, which were selected for further analysis in the present study (Supplemental Table 1). We then expressed $V_H H$-hFc fusion proteins to measure their antigen-binding avidity ($K_D$) (Fig. 2c). B12 (1.2 nM) and G8 (3.1 nM) showed stronger binding avidity to 4Ig than C4 (21 nM). G8 (0.1 nM) exhibited sub-nanomolar binding avidity to 2Ig, followed by B12 (1.3 nM), with C4 having a modest binding avidity. Moreover, we performed western blot and protein ELISA to validate the interaction between nanobodies and 4Ig or 2Ig. We also included 376.96, a mouse-derived anti-human B7-H3 mAb which is used in current clinical trials[28]. As shown in Fig. 2d, e, both 376.96 and G8 recognized 4Ig and 2Ig, whereas B12 and C4 captured only 4Ig. C4 recognized PNGase F-digested 2Ig rather than natural 2Ig, indicating that the N-linked asparagine affects the C4-2Ig interaction. Furthermore, a cell surface antigen-binding assay showed specific binding of all three $V_H H$-Fc (C4, B12, and G8) to B7-H3⁺ tumor cell lines but not to B7-H3⁻ cells (Fig. 2f). By comparison, the binding ability of B12 and G8 was more robust than two well-known anti-B7-H3 mAbs, 376.96 and MGA271[29]. Taken together, we isolated three high-affinity nanobodies that bind native human 4IgB7-H3 on tumor cells.

### Dromedary camel nanobodies recognize unique epitopes in B7-H3

The antigen-binding epitope is crucial to the biological function of therapeutic antibodies and antibody-based CAR-T cells[21]. We first performed a cross-competition assay on the Octet platform and found that G8 and 376.96 competitively bind to the antigen, suggesting their binding epitopes overlap (Fig. 3a). By contrast, C4 and B12 bind unique epitopes. Epitope mapping ELISA was further performed based on an array of 4Ig-derived peptides (Fig. 3b). Only G8 and 376.96 showed significant binding potency to peptide #10 (QRVRVADEG**S**FTCFVSIR) and the adjacent peptide #11 (SFTCFVSIRDFGSAAVSL) compared to C4 or B12 (Fig. 3b). These findings suggest that G8 and 376.96 have overlapping linearized epitopes located in the IgV domains of 4Ig, whereas C4 and B12 have unique conformational epitopes (Supplementary Fig. 2).

To further determine the epitope of C4 and B12, we designed eight truncated 4IgB7-H3-rFc-His, including single $IgV_1$, $IgV_2$, $IgC_1$, and $IgC_2$ domains, C-terminus ($IgC_1$-$V_2$-$C_2$), N-terminus ($IgV_1$-$C_1$-$V_2$), $IgV_1$-$C_1$ ($IgV_1 + IgC_1$), and $IgV_2$-$C_2$ ($IgV_2 + IgC_2$) (Fig. 3c). Each fragment was produced in mammalian cells and validated by anti-His antibody (Fig. 3c). We subsequently examined the interaction of purified fragment with $V_H H$-Fc. As shown in Fig. 3d and Supplementary Fig. 3, B12 and C4 bind to IgC ($IgC_1$ and $IgC_2$) rather than IgV ($IgV_1$ and $IgV_2$), while G8 and 376.96 bind to the IgV domains. All nanobodies interacted with C-terminus, N-terminus, $IgV_1 + C_1$, and $IgV_2 + C_2$ fragments that contain both IgV and IgC domains. In addition, these results were visualized by

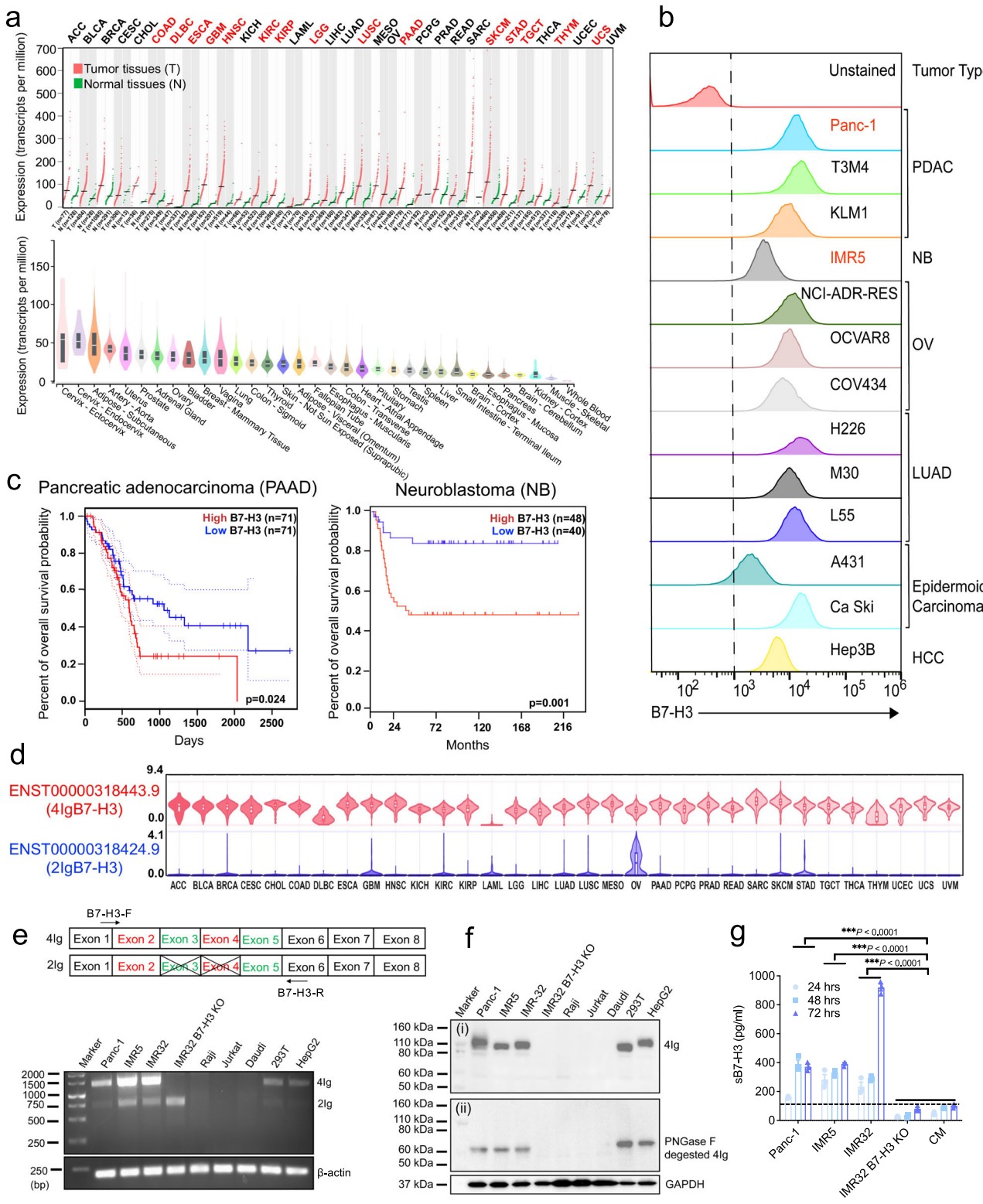

computational docking (see Methods), showing that B12 and C4 bind to distinct regions of IgC domains while the epitopes of G8 and 376.96 overlap (Fig. 3e).

## B7-H3 nanobody-based CAR-T cells lyse human PDAC and NB tumor cells in vitro

To generate B7-H3 CAR-T cells using the above nanobodies, we inserted individual V$_H$H fragment into our CAR vector that is being used in the clinic (NCT05003895)[21] (Fig. 4a). The human truncated

epidermal growth factor receptor (hEGFRt) was expressed separately as the cell surface marker for examining CAR transduction. We also produced 376.96(scFv)-CAR-T cells. The transduction efficiency of nanobody-based CAR-T cells was over 60%, slightly higher than 376.96(scFv)-CAR-T (52.9%) (Supplementary Fig. 4A). Compared to the control CD19 CAR-T cells that bind an irrelevant antigen, the B7-H3 CAR-T cells significantly lysed four NB cell lines (IMR32, IMR5, NBEB, and LAN-1), two PDAC cell lines (Panc-1 and BxPC-3), a triple-negative breast cancer (TNBC) cell line (MDA-MB-231), and a lung

**Fig. 1 | 4IgB7-H3 is a highly overexpressed dominant isoform in multiple solid cancer types. a** B7-H3 transcripts in tumors and paired normal tissues (dot plot) were analyzed using Gene Expression Profiling Interactive Analysis 2 (GEPIA 2)[55]. Red indicates significant upregulation and black shows no difference. The X-axis displays tumor samples (T) in red [log2 (TPM + 1)] and normal samples (N) in green. Four-way ANOVA and LIMMA. The over-expressed gene was defined as $\log_2 FC$ [median (tumor)-median (normal)] > 1, percentage > 0.9. The violin plot from Genotype-Tissue Expression (GTEx)[56] depicts B7-H3 expression in diverse human normal tissues. Data were presented as median (mid-line in the plot), [25 percentile (lower bound in the plot), 75 percentile (upper bound in the plot)]. $n = 3–1085$ individual tissues/group. **b** Cell surface B7-H3 expression in multiple human solid tumor cell lines. **c** Kaplan-Meier analysis of overall survival in patients with PDAC and NB with high and low B7-H3 expression from the GEPIA 2 (TCGA and GTEx datasets) and R2 Genomics Analysis and Visualization datasets (TCGA datasets). $*P = 0.024$, $**P < 0.001$, Log-rank test. **d** Distribution boxplots of 4Ig and 2Ig isoforms in various cancer types from GEPIA 2. Data were presented as median (mid-line in the plot), [25 percentile (lower bound in the plot), 75 percentile (upper bound in the plot)] of the log2-transformed normalized B7-H3 expression. $n = 26945$ independent samples. **e** Schematic design of a pair of B7-H3-specific primers to distinguish 4Ig and 2Ig isoforms. Reverse transcription-polymerase chain reaction (RT-PCR) reaction of two isoforms and housekeeping gene β-actin in multiple tumor cell lines. 2Ig (2IgB7-H3); 4Ig (4IgB7-H3). $n = 4$ independent experiments. **f** Western blot detects B7-H3 protein in tumor cell lysates with or without PNGase F digestion using anti-B7-H3 mAb (D9M2L). $n = 4$ independent experiments. **g** Concentration of soluble B7-H3 (sB7-H3) in the cultured tumor cells supernatant over a time course. RPMI culture media (CM) and IMR32 B7-H3 KO cell supernatant represent the negative control. $n = 3$ independent experiments. $***P < 0.001$, two-tailed unpaired Student's $t$ test. Values represent mean ± SEM. Source data is provided as a Source Data file.

adenocarcinoma (LUAD) cell line (H226), but not IMR32 B7-H3 KO cells, suggesting that their cytotoxicity is antigen-dependent (Fig. 4b, c and Supplementary Fig. 4B). Among the three healthy donors-derived B7-H3 CAR-T cells used in our study, we noticed the variation in the frequencies of CD4$^+$ (41.4-74%) and CD8$^+$ (25.1-62.8%) T-cell sub-populations. However, our findings consistently demonstrated a markedly higher level of cytotoxicity exhibited by B7-H3 CAR-T cells compared to the control group when co-incubated with tumor cell lines (Supplementary Fig. 4C). Although nanobody-based CAR-T cells and 376.96 scFv-based CAR-T cells have comparable cytotoxicity against IMR5 tumor cells, a significantly higher level of cytokines (IFN-γ, IL-2, and TNF-α) was released from B12(V$_H$H)-CAR-T cells compared to others (Fig. 4c, d).

To analyze the underlying mechanisms of CAR-T activity, we first measured the antigen-binding capacity of B7-H3 CAR-T cells. We found that the binding capacity of B12(V$_H$H)-CAR-T cells was the most potent for either 10 or 1 μg/ml 4Ig protein (Fig. 4e). For 2Ig, all different B7-H3 CAR-T cells still showed considerable binding ability to 10 μg/ml protein, whereas their binding to 1 μg/ml protein was noticeably weaker. To further evaluate the impact of 2Ig antigen in B12(V$_H$H)-CAR-T cells function, we implemented a functional blocking assay by adding recombinant 4Ig or 2Ig protein into the CAR-T-tumor cells incubation system. We found that only 4Ig protein could block the cytotoxicity of B12(V$_H$H)-CAR-T cells in a dose-dependent 48- or 72-hour incubation (Fig. 4f). These findings suggested that native 2Ig antigen might not be involved in the functional activation of B12(V$_H$H)-CAR-T cells.

The enhancement of T cell function is driven by two transcriptional modulators, nuclear factor-κB (NF-κB) and nuclear factor of activated T cells (NFAT)[30]. To evaluate the potential of B7-H3 CAR activation, we established CAR-Jurkat NF-κB and NFAT reporter cell lines that generate an enhanced tdTomato response to B7-H3 stimulation (see Methods). A high-level tdTomato signal in live B12(V$_H$H)-CAR-Jurkat reporter cells was observed following 5 hours of incubation with GFP overexpressing Panc-1 tumor cells (Fig. 4g). We then performed the Jurkat reporter assay involving different B7-H3 CARs and found that B12(V$_H$H)-CAR was most effective in activating NF-κB and NFAT signaling (Fig. 4h, i).

Overall, we demonstrated that B7-H3 nanobody-based CAR-T cells were functional, and B12(V$_H$H)-CAR-T cells might be more potent due to their high antigen-binding capacity and more robust T-cell activation.

## B7-H3 nanobody-based CAR-T cells eradicate pancreatic cancer in mice

To preclinically evaluate the antitumor efficacy of B7-H3 nanobody-based CAR-T cells, we constructed a metastatic PDAC mouse model via intravenous (i.v.) injection of human Panc-1 cells in immunodeficient mice (NSG, NOD/*SCID/IL2rγ*$^{-/-}$), and twenty days later treated mice with a single injection of 10 million human CAR-T cells (Fig. 5a). During three weeks of infusion, metastatic tumors grew progressively in CD19 CAR-T cells control mice, while B7-H3 CAR-T cells significantly eliminated tumor burden (Fig. 5b, c). Crucially, B12(V$_H$H)-CAR-T cells and C4(V$_H$H)-CAR-T cells exhibited comparable antitumor efficacy, shrinking tumors faster than G8(V$_H$H)-CAR-T cells. To compare the antitumor activity between B12(V$_H$H)-CAR-T and C4(V$_H$H)-CAR-T, we then treated Panc-1 tumor-bearing mice with 5 million human CAR-T cells followed by tumor rechallenge (Fig. 5d). While tumors in the CD19 CAR-T control group grew progressively, mice infused with either B12(V$_H$H)-CAR-T or C4(V$_H$H)-CAR-T cells showed rapid tumor-shrinkage until mice were eventually tumor-free (Fig. 5e, f). A secondary dose of Panc-1 cells was implanted in these tumor-free mice and the control group. 100% of C4(V$_H$H)-CAR-T mice and 60% of B12(V$_H$H)-CAR-T mice remained tumor-free, leading to a 100% survival advantage compared with control mice (Fig. 5g). In the CD19 CAR-T mouse (ID #139), the Panc-1 tumors invasively grow in the whole lung, partial liver, pancreas, and kidney, whereas no tumor nodules were observed in the B12(V$_H$H)-CAR-T mouse (#137) nor C4(V$_H$H)-CAR-T mouse (#153) at week 10, suggesting potent persistence CAR-T activity in vivo (Fig. 5h). However, the B12(V$_H$H) CAR-T mouse (#155) had a tumor relapse upon rechallenge, with tiny tumor nodules found in the murine liver and kidney.

Upon further dose reduction from 5 to 2.5 million CAR-T cells, B12(V$_H$H)-CAR-T cells and C4(V$_H$H)-CAR-T cells maintained superior antitumor efficacy against Panc-1 compared to 376.96(scFv)-CAR-T cells and G8(V$_H$H)-CAR-T cells, resulting in longer survival time (Fig. 5i–k and Supplementary Fig. 5A). To identify the factors contributing to the high efficiency of persistent B12(V$_H$H)-CAR-T cells and C4(V$_H$H)-CAR-T cells, we analyzed circulating CAR-T cells at three different time points post-infusion. An increased absolute number of B12(V$_H$H)-CAR-T cells and C4(V$_H$H)-CAR-T cells with lower PD-1 expression than the other two constructs was found during weeks 3–4 (Fig. 5l, m). We also examined the phenotypes of circulating CAR-T cells and found a higher percentage of central memory T cells (Tcm) and effector memory T cells (Tem) proportions in the persistent CD8$^+$ for all nanobody-based CAR-T cells compared to 376.96(scFv)-CAR-T cells (Fig. 5n). Correspondingly, in the mice treated with 2.5 million CAR-T cells, G8(V$_H$H)-CAR-T cells showed a lower percentage of Tcm than the other three constructs, which was correlated with less efficacy of G8(V$_H$H)-CAR-T cells (Supplementary Fig. 5B). Finally, we established an orthotopic pancreatic tumor mouse model and demonstrated that treatment of 2.5 million B12(V$_H$H)-CAR-T cells could significantly regress a Panc-1 orthotopic tumors (Figs. 5o, 5p and Supplementary Fig. 6). These findings further validate that B7-H3 IgC-targeted B12(V$_H$H)-based CAR-T cells have the most potent antitumor activity in solid tumors.

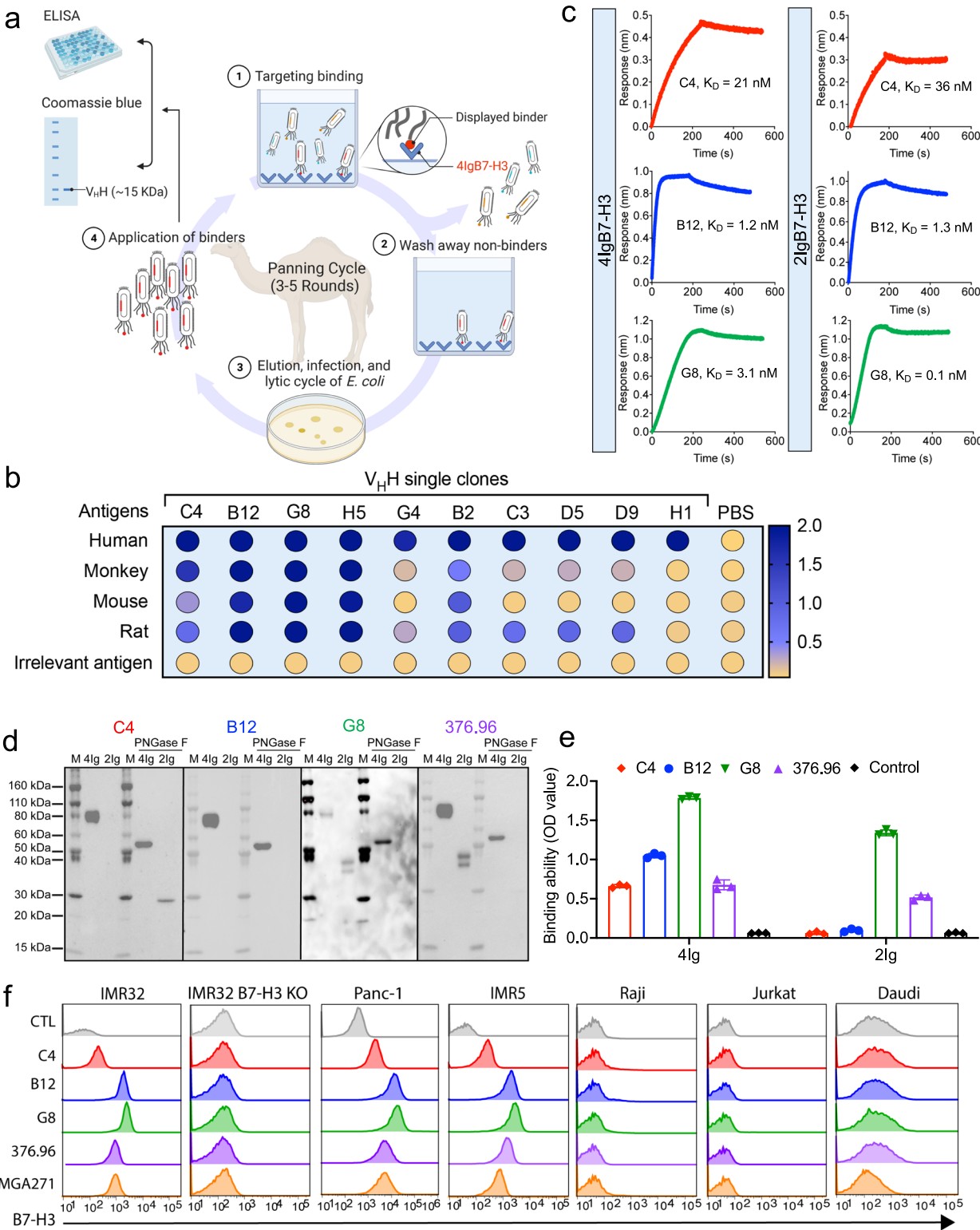

**Fig. 2 | Isolation of high-affinity camel nanobody ($V_HH$) against 4IgB7-H3.**
**a** Schematic showing phage panning (created with BioRender.com). The B7-H3-targeted phage binders were isolated from 3–5 rounds of phage panning in eight individual camel phage libraries and further validated by phage ELISA and protein production. **b** Monoclonal phage ELISA to examine the cross-species reactivity of ten individual anti-B7-H3 phage clones to the antigen of humans, monkeys, mice, and rats. An irrelevant antigen BSA was used as the control. $n = 3$ independent experiments. **c** The antigen binding kinetics of $V_HH$-Fc (C4, B12, and G8) to 4Ig or 2Ig proteins via the Octet platform. $n = 3$ independent experiments. **d** The interaction between $V_HH$-Fc and 4Ig or 2Ig proteins (with or without PNGase F digestion) via western blot. The anti-B7-H3 mAb 376.96 was used as a control. $n = 4$ independent experiment. **e** ELISA to examine the interaction between $V_HH$-Fc and 4Ig or 2Ig proteins. $n = 3$ independent experiments. Values represent mean ± SEM. **f** Flow cytometry was performed to monitor the binding capacity of $V_HH$-Fc to cell surface B7-H3 expressed in varying human tumor cell lines. Two anti-B7-H3 mAbs, 376.96 and MGA271, were used as control. $n = 4$ independent experiments. Source data is provided as a Source Data file.

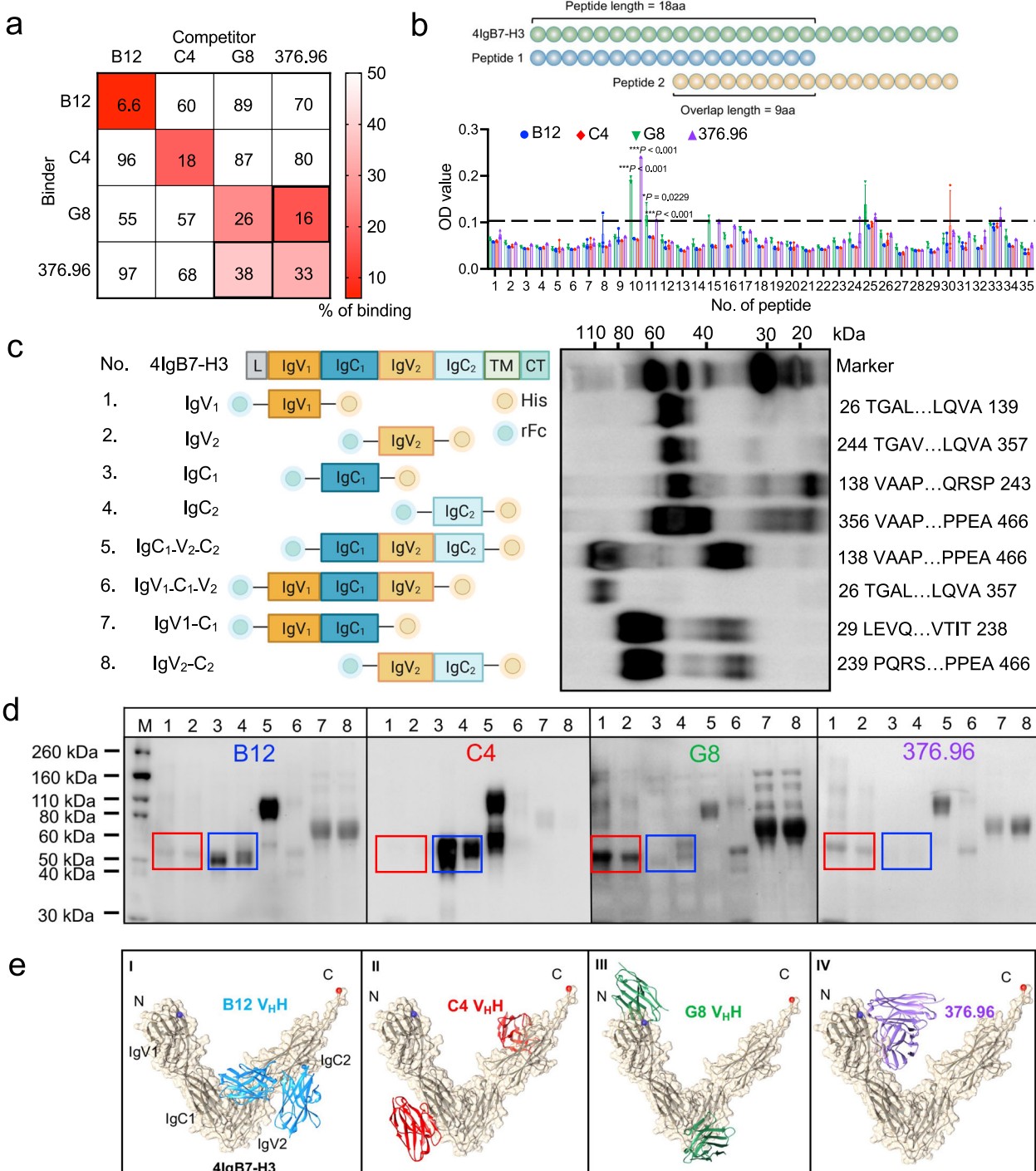

**Fig. 3 | Epitope prediction of anti-B7-H3 nanobodies. a** Cross-competition assay of V$_H$H-Fc and 376.96 mAb to the 4Ig antigen on Octet. **b** Epitope mapping of individual V$_H$H-Fc and 376.96. Schematic design for constructing a peptide array of 4Ig (total of 35 peptides). Each peptide is 18 amino acids (aa) in length, and every two junctions overlap by 9 aa. ELISA was performed to examine the binding epitope of antibodies. The baseline (dotted line) was set up at OD = 0.11 based on the binding signal. n = 3 independent experiments. Values represent mean ± SEM. ***P < 0.001, *P = 0.0229, two-tailed unpaired Student's t test. **c** Diagram of the truncated 4Ig fragments, with N-terminal rabbit Fc (rFc) and C-terminal His tag, is

indicated on the left side, and the results of western blot using anti-His-HRP antibody are shown on the right side. The truncated amino acid position, which is based on the full-length 4IgB7-H3, is shown as a number in the diagram of domain structure. n = 3 independent experiments. **d** Interaction between V$_H$H-Fc (or 376.96) and individual 4Ig fragment by western blot. The recognition of antibodies to IgV$_1$ (No. 1) and IgV$_2$ (No. 2) was in a red rectangle, and the blue rectangle highlighted IgC$_1$ (No. 3) and IgC$_2$ (No. 4). n = 3 independent experiments. **e** The complex structure models of 4IgB7-H3 and V$_H$H (B12, C4, G8) or scFv of 376.96. Source data is provided as a Source Data file.

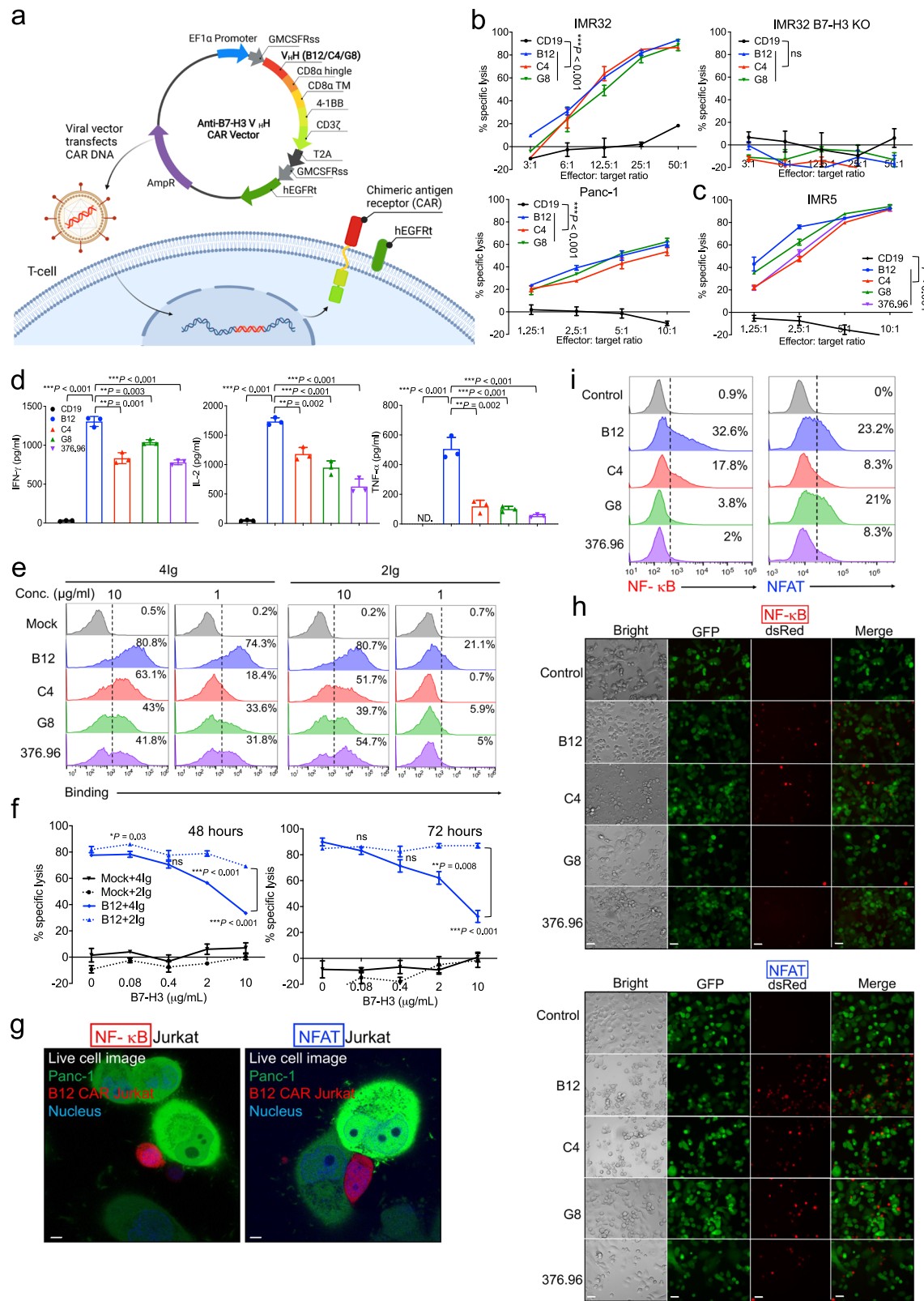

Taken together, these findings demonstrated that B12($V_H$H)-CAR-T cells and C4($V_H$H)-CAR-T cells targeting the B7-H3 IgC domain have persistent superior antitumor activity compared to IgV-targeted 376.96(scFv)-CAR-T cells and G8($V_H$H)-CAR-T cells against pancreatic tumors in mice. The percentage of Tcm and Tem subsets might be positively related to the persistent antitumor efficiency of CAR-T cells in vivo.

## B12($V_H$H)-CAR-T cells have the most potent antitumor activity against metastatic and large solid tumors in mice

To examine our nanobody-based CAR-T cells in neuroblastoma, we established a metastatic neuroblastoma mouse model by i.v. injection of human IMR5 cells (Fig. 6a). Thirty-five days later, tumors metastatically grew in the mouse abdominal cavity, liver, legs, and spine, which are rich in nerves (Fig. 6b). In comparison to CD19 CAR-T cells,

**Fig. 4 | B7-H3 nanobody-based CAR-T cells effectively lysed PADC and NB cells in vitro. a** Schematic showing B7-H3 CAR-T design and production (created with BioRender.com). Lentiviral vectors encode CAR and hEGFRt via a T2A ribosomal skipping sequence, leading to CAR expression on the T cell surface. **b** Cytolytic activity of B7-H3 CAR-T and CD19 CAR-T against IMR32, IMR32 B7-H3 KO, and Panc-1 cells, at various E/T ratios for 24 hours incubation. $n = 3$ independent experiments. Values represent mean ± SEM. ***$P < 0.001$, ns $P > 0.05$, two-tailed unpaired Student's $t$ test. **c** Cytolytic activity of B7-H3 CAR-T cells against IMR5 cells at various E/T ratios for 24 hours of incubation. 376.96 represent 376.96(scFv)-CAR-T cells. $n = 3$ independent experiments. Values represent mean ± SEM. ***$P < 0.001$, two-tailed unpaired Student's $t$ test. **d** B7-H3 CAR-T cytokine release (IFN-γ, IL-2, and TNF-α) upon co-culture with IMR5 cells. $n = 3$ independent experiments. Values represent mean ± SEM. **$P < 0.01$, ***$P < 0.001$, two-tailed unpaired Student's $t$ test. **e** B7-H3 CAR-T binding to recombinant human 4Ig and 2Ig protein. $n = 3$ independent experiments. **f** 4Ig protein inhibited B12($V_H$H)-CAR-T killing on Panc-1 cells for 48 hours and 72 hours incubation. ns, not significant. $n = 3$ independent experiments. Values represent mean ± SEM. *$P = 0.03$, **$P = 0.08$, ***$P < 0.001$, ns $P > 0.05$, two-tailed unpaired Student's $t$ test. **g** Live cell imaging of B12($V_H$H)-CAR-Jurkat NF-κB/NFAT reporter tdTomato cells (dsred) and GFP (green) overexpressing Panc-1 cells by confocal microscopy at five hours incubation. The nucleus was labeled by Hoechst (blue). Scale bar, 5 μm. $n = 3$ independent experiments. **h** Confocal microscopy of CAR-Jurkat reporter cells upon interacting with Panc-1 GL cells at five hours incubation. Cell images at 20x magnification. Scale bar, 50 μm. $n = 3$ independent experiments. **i** Quantitative percentage of activated CAR-Jurkat reporter cells by flow cytometry. All stimulated CAR-Jurkat reporter cells from each whole well were harvested for flow cytometry. Three biological replicate experiments were performed. CAR-T cells were from donor #076's PBMC. Source data is provided as a Source Data file.

B12($V_H$H)-CAR-T cells significantly inhibited tumor growth, followed by C4($V_H$H)-CAR-T cells, whereas G8($V_H$H)-CAR-T cells and 376.96(scFv)-CAR-T cells were ineffective (Fig. 6c, d). Correspondingly, significantly reduced plasma sB7-H3 and prolonged overall survival were observed in mice treated with B12($V_H$H)-CAR-T cells and C4($V_H$H)-CAR-T cells (Fig. 6e, f).

In the clinic, cancer patients usually have very large tumors that are difficult to treat using existing drugs. Since we found our B7-H3 nanobody CAR-T cells were very active, we decided to explore the potential of these CAR-T cells in treating large tumors. To this end, we developed another neuroblastoma xenograft mouse model by subcutaneously (s.c.) implanting human NBEB tumor cells into NSG mice and growing the tumors to a large size (400-800 mm³) (Fig. 6g). This aggressive tumor in CD19 CAR-T treated control mice increased approximately 10-fold in size in eight days (from 400 mm³ to 4000 mm³) (Fig. 6h). Strikingly, only B12($V_H$H)-CAR-T cells significantly controlled tumor growth, whereas C4($V_H$H)-CAR-T cells and 376.96(scFv)-CAR-T cells showed limited efficacy (Fig. 6h), which was visualized by the end of this study (Supplementary Fig. 7). In contrast, NBEB B7-H3 KO tumors were not inhibited by B12($V_H$H)-CAR-T cells (Supplementary Fig. 8). The memory phenotypes of circulating CAR-T cells were also analyzed by flow cytometry. The circulating CD8+ B12($V_H$H)-CAR-T cells comprised a higher percentage of the Tem subpopulation than other treatment groups (Fig. 6i). In contrast, CD19 CAR-T cells showed low Tem frequency in both CD4+ T and CD8+ T subsets, suggesting that the occupation of the Tem subpopulation was crucial to CAR-T cells' high-efficient antitumor activity. We performed histopathological analyses of NBEB tumors on day 9 after CAR-T cell infusion. B12($V_H$H)-CAR-T cells induced significantly more tumor-infiltrating lymphocytes (TILs) as compared with C4($V_H$H)-CAR-T cells, 376.96(scFv)-CAR-T cells, and irrelevant CD19 CAR-T cells (Fig. 6j and Supplementary Fig. 9). Increased apoptotic cells were observed in B12($V_H$H)-CAR-T rich areas (Supplementary Fig. 10). Given the decreased tumor size, increased TIL density in tumors, and the spatial colocalization of increased T-cells with increased apoptosis in B12($V_H$H)-CAR-T treated tumors, we conclude that B12($V_H$H)-CAR-T cells mediated tumor cell killing.

To enable a parallel comparison, we established an additional PDAC mouse model by injecting human BxPC-3 cells intraperitoneally (i.p.), followed by infusion of 5 million CAR-T cells (Fig. 6k). Tumors were allowed to grow to a large size (average tumor signal: $5 \times 10^{10}$ p/sec/cm²/sr, equivalent to around 250 mm³ tumors) before CAR-T treatment, revealing superior antitumor efficacy of B12($V_H$H)-CAR-T cells, followed by C4($V_H$H)-CAR-T cells. In contrast, 376.96($V_H$H)-CAR-T cells showed limited efficacy (Fig. 6l, m).

### Persistent single B12($V_H$H)-CAR-T cell is polyfunctional
Polyfunctional T cells, having the ability to produce multiple cytokines and chemokines, provide a more effective immune response than cells that secrete only a single cytokine[31,32]. We analyzed single-cell-based polyfunctionality to understand the heterogeneous functional activation of our most effective B12($V_H$H)-CAR-T cells upon tumor challenge in vivo. To do this, due to the undetectable tumors in B12($V_H$H)-CAR-T treated IMR5 tumor-bearing mice, we isolated long-acting B7-H3 CAR-T cells from the spleens at week 5 after CAR-T cell infusion, followed by ex vivo tumor (IMR32 and IMR32 B7-H3 KO) incubation, and a single-cell-based 32-plex cytokines/chemokines microfluidics device was subsequently performed (Fig. 7a). We found that the mouse spleen-isolated CD4+CAR-T cells were the dominant subpopulation as compared with CD8+CAR-T cells (84%±3.0 versus 11%±1.0), which allowed us to focus on the CD4+CAR-T cell subset. The single-cell analysis showed a cluster of polyfunctional B12($V_H$H)-CAR-T cells within the mixture of distinct cell subsets, and this population secreted cytokines belonging to the anti-tumor effector proteins (Fig. 7b and Supplementary Fig. 11). We quantified the percentage of polyfunctional subsets across different B7-H3 CAR-T cells and found that only CD4+ B12($V_H$H)-CAR-T cells have a small percentage of polyfunctional subpopulation upon IMR32 cells stimulation compared with IMR32 B7-H3 KO cells (Fig. 7c). The breadth of immune responses includes anti-tumor effectors (Granzyme B, IFN-γ, MIP-1α, TNF-α), stimulatory (GM-CSF, IL-8), and chemoattractive (IP-10 and MIP-1β) factors. Due to limited cell numbers, a substantial population of polyfunctional CD8+ CAR-T cells was not observed (Supplementary Fig. 12).

### Single-cell proteomics and transcriptomics expression profiles between high and low polyfunctional clusters
To better understand the molecular changes of persistent CAR-T cells in mice, we performed new single-cell assay (See Methods) that could simultaneously capture proteins and RNA transcripts in the same single cells (Fig. 7d). We discovered that the 5-week persistent B12($V_H$H)-CAR-T cells isolated from IMR5 mice could be clustered into two groups with either high or low polyfunctionality (72 versus 23 single CAR-T cells) after incubation with IMR5 tumor cells in vitro (Fig. 7d). Multiple proteins were found mainly expressed in the high polyfunctionality CAR-T cells rather than low polyfunctionality single cells, including GM-CSF, Granzyme B, IFN-γ, IL-2, IL-8, MIP-1β, Perforin, TNF-α, p-IκBα, p-MEK1/2, p-NF-κB p65, p-Stat1, and p-Stat5 in response to B7-H3 antigen (Fig. 7e). The transcriptomic profiles coupled with the same single B12($V_H$H)-CAR-T cells allowed us to identify 32 genes with statistically significant expression differences (p < 0.05) between the high and low polyfunctionality subsets with the top 5 upregulated genes (*EPRS1*, *ATP5PB*, *TUBA1C*, *RBM39*, and *EIF1AX*) in the high polyfunctionality cluster (Fig. 7e and Supplementary Table 2). Reactome analysis indicated that the high polyfunctionality group contained genes associated with translation, formation of the cornified envelope,

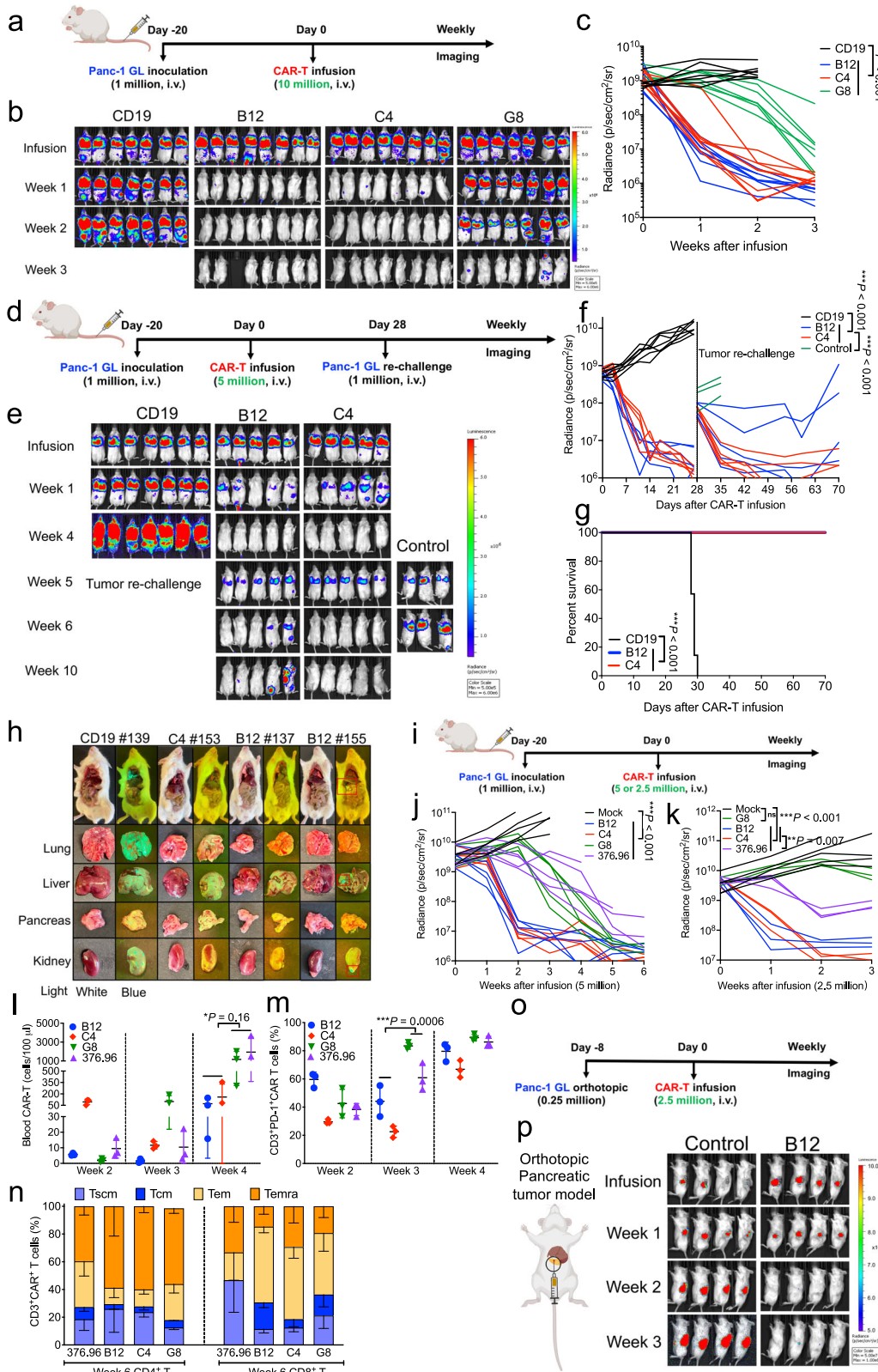

post-transcriptional silencing, and competition for endogenous RNAs (ceRNAs) that regulate translation of phosphatase and tensin homolog deleted on chromosome 10 (PTEN) (Fig. 7f and Supplementary Table 3). Our data show that CAR-T cell activity is regulated by transcriptional and translational responses to tumors and indicate that a cluster of elevated gene expression might be important for the persistence of functional CAR-T cells.

## Discussion

In the present study, we analyzed the isoforms of B7-H3 at the RNA and protein levels and validated that the 4Ig isoform is a therapeutic target in tumors. One of the major obstacles in CAR generation is the optimization of antigen recognition domains. The location of an antibody binding site on the target is critical for the efficacy of CARs. However, it is generally unknown what epitope works best for a cancer target.

**Fig. 5 | B7-H3 nanobody-based CAR-T cells exhibit superior persistent anti-tumor activity against human pancreatic cancer in mice. a** Experimental schema of the Panc-1 metastasis xenograft mouse model (created with BioRender.com). *n* = 7/8 mice/group. **b, c** Representative tumor bioluminescence images of mice **b** and tumor growth curve **c** measured by bioluminescence. *n* = 7/8 mice/group. ***P* < 0.001, two-tailed unpaired Student's *t* test. **d** Experimental schema of the Panc-1 metastasis xenograft mouse model under tumor-rechallenge (created with BioRender.com). *n* = 5/7 mice/group. Three mice were used as the control. **e, f** Representative tumor bioluminescence images of mice **e** and tumor growth curve **f**. *n* = 5/7 mice/group. ***P* < 0.001, two-tailed unpaired Student's *t* test. **g** Kaplan-Meier survival curve post-CAR-T cells infusion. ***P* < 0.001, Log-rank test. **h** Tumor metastasis images in mice infused with CD19 or B7-H3 CAR-T cells. Mouse tissues and tumors (GFP) were visualized by the white and UV/blue light, respectively. **i** Dose-dependent effects of B7-H3 CAR-T cells in the Panc-1 metastasis mouse model (created with BioRender.com). *n* = 3/5 mice/group. **j, k** Tumor bioluminescence growth curve with 5 million (**j**) or 2.5 million (**k**) B7-H3 CAR-T cells. ***P* = 0.007, ****P* < 0.001, two-tailed unpaired Student's *t* test. **l** Absolute CAR-T count from 100 μl blood was identified in mouse peripheral blood after week 2/3/4 of infusion. *n* = 3 individual samples/group. Values represent mean ± SEM. **P* = 0.16, two-tailed unpaired Student's *t* test. **m** PD-1 expression in circulated CD3+CAR+ cells at week 2/3/4 of infusion. *n* = 3 individual samples/group. Values represent mean ± SEM. ***P* = 0.0006, two-tailed unpaired Student's *t* test. **n** Distribution of memory subsets in CD4+/CD8+ CAR+-T cell subpopulations at week 6 of infusion. *n* = 3 individual samples/group. Values represent mean ± SEM. **o** Experimental schema of the Panc-1 orthotopic mouse model (created with BioRender.com). *n* = 4 mice/group. **p** Tumor bioluminescence images of orthotopic Panc-1 mice (created with BioRender.com) in the CAR-T treatment groups. CAR-T cells from donor #076's PBMC. Source data is provided as a Source Data file.

Since 4IgB7-H3 has two distinct epitope domains, IgV and IgC, while the epitope of most B7-H3 antibodies has not been described, we constructed our large dromedary camel V$_H$H phage libraries and used them to isolate a panel of nanobodies that bind various epitopes on 4IgB7-H3, including previously unreported epitopes in the IgC domain. We demonstrated that the CAR-T cells based on the nanobodies targeting particular epitopes in the IgC domain were more active than those targeting IgV against large solid tumors in mice, including previously reported scFv-based CAR-T cells. Notably, only the most potent nanobody-based CAR-T cells based on the B12 V$_H$H could inhibit two large tumor models for neuroblastoma and pancreatic cancer in mice. To analyze the functionality of nanobody-based CAR-T cells, we used single-cell transcriptome RNA sequencing coupled with single-cell functional proteomics and identified a panel of upregulated genes in the polyfunctional CAR-T cells harvested from mice. We also made a reporter assay to analyze CAR-T cells for NF-κB and NFAT signaling. Our results indicate that the activity of CAR-T cells is regulated by transcriptional, translational, and T-cell signaling responses to the tumors in an antigen-specific manner. Importantly, using large camel V$_H$H libraries to isolate nanobodies that recognize all the epitope domains on a tumor antigen, we identified a new epitope that enabled the development of nanobody-based CAR-T cells with activity against large tumors.

Humanization of nanobodies may reduce immunogenicity in humans. However, the impact of nanobody humanization on immunogenicity remains uncertain, as some V$_H$Hs show no immunogenicity while certain fully human VHs can elicit anti-drug antibodies[33]. The FDA-approved CAR-T (ciltacabtagene autoleucel) targeting BCMA for treating multiple myeloma was derived from two camelid V$_H$Hs without humanization[34]. Therefore, we consider humanization might be advantageous for the use of nanobodies in the clinic, but we also recognize that it may not always be necessary due to the uncertainty of immunogenicity and the high sequence similarity between camelid V$_H$Hs and human VHs. Given their ability to probe a wide range of surface epitopes, nanobodies could be particularly powerful in screening campaigns to identify the most potent antigenic epitopes for CAR-T development, helping improve CAR-T activity against solid tumors.

We illustrated that B7-H3 is aberrantly overexpressed across a diverse spectrum of solid tumors while present at low levels in most normal tissues, which recapitulated the B7-H3 expression in solid tumors previously reported[5,35,36]. B7-H3 has been reported to have two isoforms, 4Ig and 2Ig, and both isoforms have IgV and IgC domains. Data extracted from TCGA database and our RT-PCR results demonstrated that 4Ig has a much higher transcriptional level than 2Ig in most human malignant tissues and B7-H3+ tumor cell lines. In the present study, we developed nanobodies targeting either IgV or IgC domains of B7-H3 and demonstrated that all nanobodies could bind to both 4Ig and 2Ig protein, but more strongly to 4Ig than to 2Ig. Using a commercial anti-B7-H3 mAb, a previous study showed that the 4Ig was specific for glioblastoma, whereas 2Ig was expressed in non-cancerous brain tissue but contributed to the subsequent recurrence[26]. However, we could not validate a true "2Ig" in multiple B7-H3+ cell lysates using the same anti-B7-H3 mAb (Supplementary Fig. 1). Overall, our data show that the 4Ig, as the dominant isoform, is a target for developing immunotherapeutic strategies. B12(V$_H$H)-CAR-T cells have dramatically higher antitumor potency than other constructs in multiple solid tumor models in our study. Our findings highlight the significance of Tem and Tcm subpopulations in achieving highly efficient antitumor activity with persistent circulated CAR-T cells. Tem cells have enhanced tissue migration receptors and immediate effector functions, while Tcm-derived CD8 + CAR T cells exhibit superior in vivo survival and proliferation compared to other subsets[37,38]. Interestingly, a main percentage of CD4+ rather than CD8+ CAR-T cell population was found in the mouse spleen-isolated long-term persistent CAR-T cells. Melenhorst JJ et al. studied long-lasting CD19 CAR-T cells in chronic lymphocytic leukemia patients who achieved complete remission[39]. They found that the CAR-T cells persisted for over ten years, with CD4+ CAR-T cells becoming the dominant population over time while maintaining ongoing functional activation and proliferation, indicating that the biological significance of CD4+ CAR-T cells within the tumor microenvironment might be important to understand for further study. Future research may also evaluate the potency of our B7-H3 CAR-T cells in the tumor microenvironment using immunocompetent mouse tumor models, such as KPC mouse models for pancreatic cancer[40] and TH-MYCN transgenic murine model for neuroblastoma[41].

To address the underlying functional attributes of persistent B7-H3 CAR-T cells upon antigen re-interaction, we used a single-cell platform that integrates transcriptome RNA sequencing with functional proteomics from the same CAR-T cells. Notably, besides cytokines, we observed increased expression of p-NF-κB p65 protein in the high-polyfunctional B12(V$_H$H)-CAR-T cells, consistent with our observations in CAR-Jurkat NF-κB reporter assay, indicating the effective activation of NF-κB signaling in high-polyfunctional B12(V$_H$H)-CAR T cells. In mice bearing tumors, high polyfunctional B12(V$_H$H)-CAR-T cells showed upregulated transcripts related to translation and protein synthesis. The protein encoded by the top gene *EPRS1* plays an essential role during protein synthesis and metabolism via phosphorylation upon IFN-γ stimulation. T-cell activation and differentiation involve significant changes in cellular metabolic programs, serving as a "metabolic checkpoint" that coordinates metabolic status with cellular signaling within the tumor microenvironment[42]. Additionally, other up-regulated transcripts in the high-polyfunctional CAR-T, such as *MRPL52*, *EIF1AX*, *ATP5PB*, *TUBA1C*, *CBWD1*, and *MFSD4B*, are related to the T-cell metabolic pathway. For example, *ATP5PB* encodes a subunit of mitochondrial ATP synthase, which may contribute to rapid ATP formation during T cell activation[43]; *TUBA1C* is closely correlated to the cell cycle, suggesting that high polyfunctional CAR-T cells may be proliferating more as compared to low polyfunctional subpopulation.

In summary, we isolated B7-H3 nanobodies and demonstrated that the CAR-T cells based on the B12 V$_H$H nanobody recognizing

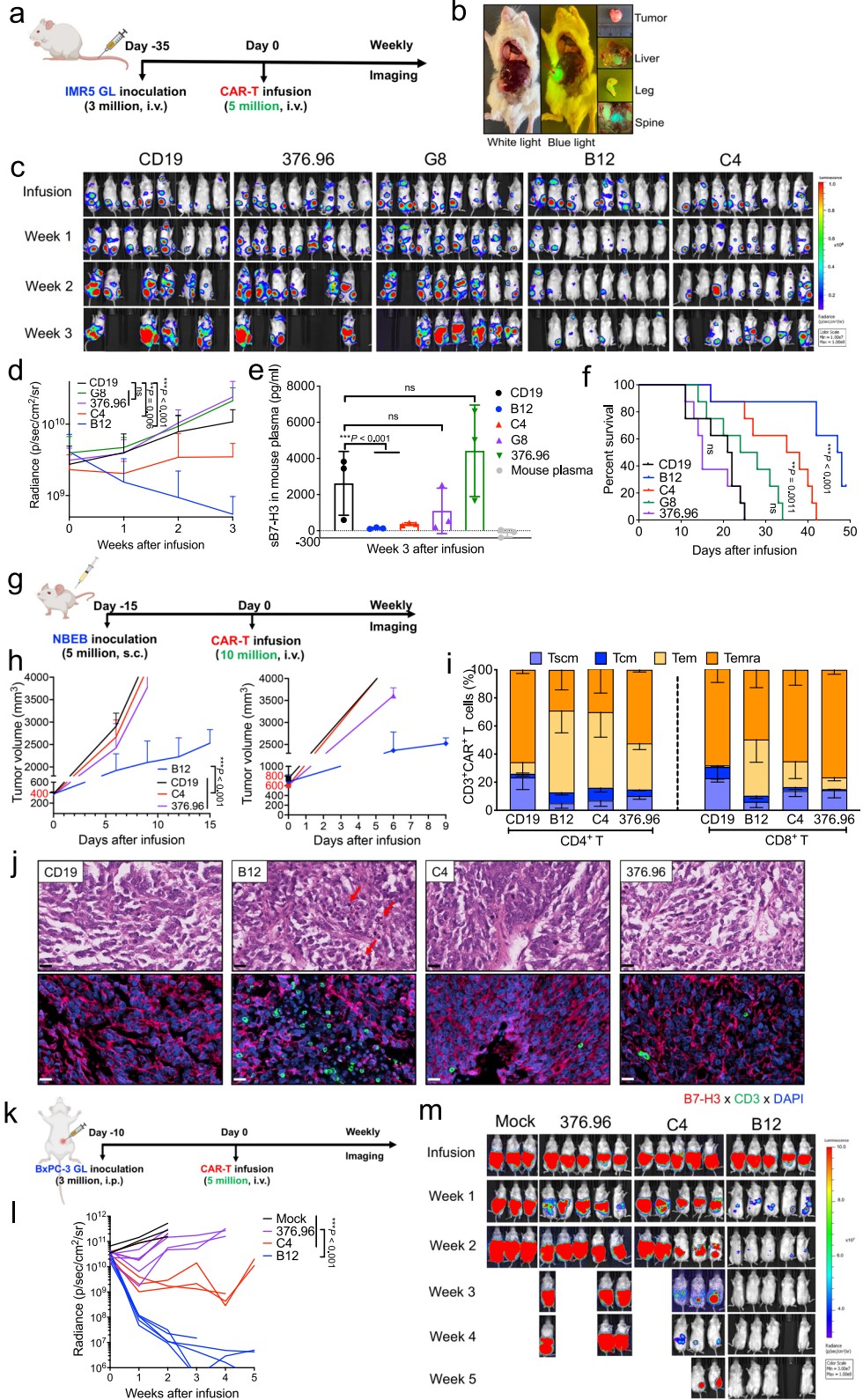

a unique epitope in the IgC domain of 4IgB7-H3 had the promising antitumor effect against advanced metastatic and large tumors in mice. This study demonstrates the potential of V$_H$H-based CAR-T cells for treating solid tumors including pancreatic cancer and neuroblastoma and provides a strategy for using nanobodies targeting rare and potent epitopes in engineering CAR-T cells.

## Methods

### Use of mice and use of human specimens in experiments

All experiments involving animals were approved by the Institutional Animal Care and Use Committee at the NIH (protocol LMB-059). Mice were co-housed under a standard 12:12 hours light/dark cycle with standard food and water (barrier, specific pathogen-free/SPF).

**Fig. 6 | B12(V_HH)-CAR-T cells inhibited metastatic and large tumor growth in xenografts. a** Experimental schema of IMR5 metastasis xenograft mouse model (created with BioRender.com). $n = 8$ mice/group. **b** Representative IMR5 tumor image. **(c-d)** Tumor bioluminescence images **c** and tumor growth curve **d** for CAR-T treatment groups. $n = 8$ mice/group. ns, not significant. Values represent mean ± SEM. **$P = 0.006$, ***$P < 0.001$, ns $P > 0.05$, two-tailed unpaired Student's $t$ test. **e** Plasma sB7-H3 levels in IMR5 mice after three weeks of CAR-T infusion. $n = 3$ individual samples/group. ns, not significant. Values represent mean ± SEM. ***$P < 0.001$, ns $P > 0.05$, two-tailed unpaired Student's $t$ test. **f** Kaplan−Meier survival curve of IMR5 mice post-CAR-T infusion. ***$P < 0.001$, ns $P > 0.05$, Log-rank test. **g** Experimental schema of the subcutaneous NBEB xenograft mouse model (created with BioRender.com). Two groups of mice, including one with tumor volume at ≈400 mm$^3$ ($n = 5$ mice/group) and another with tumor volume at ≈600−800 mm$^3$ ($n = 2$ mice/group). **h** The respective tumor growth curves of two groups of mice infused with CAR-T cells. $n = 5$ or 2 mice/group. Values represent mean ± SEM. ***$P < 0.001$, two-tailed unpaired Student's $t$ test. **i** CD4$^+$/CD8$^+$ CAR$^+$-T cell immunophenotypes post-infusion. $n = 3$ individual samples/group. Values represent mean ± SEM. **j** CAR-T cell infiltration in NBEB tumor tissues on day 9 post-infusion was analyzed through H&E and IF sections. Positive staining was quantified using anti-CD3 mAb (green) and anti-B7-H3 mAb (red). DAPI staining marked nuclei. Arrows indicate apoptotic cells. Scale bar, 20 μm. $n = 3$ independent experiments. **k** Experimental schema of the large BxPC-3 xenograft mouse model (created with BioRender.com). $n = 3/5$ mice/group. **l−m** Tumor growth curve (**l**) and representative tumor bioluminescence images of mice (**m**) upon CAR-T treatment. CAR-T cells from donor #076's PBMC. Statistical analyses are shown from more than three individual samples. Values represent mean ± SEM. ***$P < 0.001$, two-tailed unpaired Student's $t$ test. Source data is provided as a Source Data file.

Human peripheral blood samples from healthy donors were purchased from the Oklahoma Blood Institute (Oklahoma City, USA). The use of deidentified human specimens was determined to be the NIH Institutional Review Board (IRB) exempt. Because the specimens or data were not collected specifically for our study and no one on our study team has access to the subject identifiers linked to the specimens or data, our study is not considered as human subjects research.

## Cell culture

Pancreatic cancer cell lines including Panc-1, T3M4, KLM1, and BxPC-3 were obtained from Drs. Perwez Hussain, Udo Rudloff, and Christine Alewine at the National Cancer Institute (NCI, Bethesda, MD), respectively. Neuroblastoma cell lines including IMR5, IMR32, NBEB, and LAN-1 were obtained from NCI Pediatric Oncology Branch (Bethesda, MD). Lung cancer cell line M30 was a gift from Dr. Raffit Hassan at the NCI (Bethesda, MD). Triple-negative breast cancer cell line MDA-MB-231, ovarian cancer cell lines NCI-ADR-RES and COV434, lung cancer cell line H226, and epidermoid carcinoma cell line Ca Ski, were purchased from American Type Culture Collection (ATCC). Ovarian cancer cell line OVCAR8 was obtained from the National Cancer Institute (Development Therapeutics Program). Lung cancer cell line L55 was provided by Dr. Steven M. Albelda at the University of Pennsylvania (Philadelphia, PA). Two liver cancer (HCC) cell lines Hep3B, HepG2, an epidermoid carcinoma cell line A431, and HEK-293T were purchased from the American Type Culture Collection (Manassas, VA). Three lymphoma cell lines including Jurkat, Daudi, and Raji were gifts from Dr. Ira Pastan in NCI (Bethesda, MD). Luciferase-p2A-mCherry (ML) overexpressed NBEB, IMR32, IMR32 B7-H3 KO, and NBEB B7-H3 KO cell lines were obtained from Brad St Croix at the NCI (Frederick, MD)[5,44]. Panc-1, IMR5, BxPC-3, LAN-1, H226, and MDA-MB-231 cell lines were transduced with lentiviruses expressing firefly Luciferase-p2A-GFP (GL) in our lab. Hep3B and A431 cells were cultured in DMEM supplemented with 10% FBS, 1% L-glutamine, and 1% penicillin−streptomycin; other cell lines mentioned above were cultured in RPMI. Cells were maintained in a humidified atmosphere containing 5% CO$_2$ at 37 °C.

## Isolation of anti-B7-H3 nanobodies

We conducted phage panning following our laboratory protocol[45,46] using eight individual large dromedary camel (*Camelus dromedarius*) V_HH phage libraries[47]. Briefly, a total of 3-5 rounds of phage display were implemented on Maxisorp immune tubes (Thermo Fisher Scientific) coated with human 4IgB7-H3-Fc protein (produced in our lab) in phosphate-buffered saline (PBS). The anti-4IgB7-H3 phages were eluted and amplified. Single colonies were then picked and identified by performing phage ELISA. The antibody information is provided in Supplementary Table 4.

## ELISA

Phage ELISA was performed as previously described[48]. Briefly, pre-blocked phage supernatant was incubated with Maxisorp 96-well plates (Fisher Scientific) coated with 5 μg/ml B7-H3 proteins of different species, including human, monkey, mouse, rat, and the irrelevant antigen human IgG (obtained from Brad St Croix at the NCI) and PBS. The binding activity was initially determined with horseradish peroxidase (HRP)-conjugated mouse anti-M13 antibody (GE Healthcare, discontinued), and later the one from Sino Biological (Cat#11973-MM05T-H). For the antibody binding ability ELISA, the plate was coated with 5 μg/ml human 4IgB7-H3-his or human 2IgB7-H3-his in PBS, followed by adding 0.1 μg/ml B12-Fc, C4-Fc, G8-Fc, 376.96, or an irrelevant V_HH-Fc as the control. The binding was detected by goat anti-human IgG or goat anti-mouse IgG HRP-conjugated antibodies.

In the epitope mapping ELISA, we designed a panel of 35 peptides array based on the human 4IgB7-H3 extracellular domain amino acid sequence (Supplementary Table 5). Each synthesized peptide (GenScript) was 18 amino acids in length and had nine aa overlapped with adjacent peptides. 5 μg/ml peptides in PBS were used to coat the plates overnight. 1 μg/ml of B12, C4, G8, or 376.96 was added to the assay wells, and the binding was detected with a monoclonal anti-FLAG antibody conjugated with HRP and a goat anti-mouse IgG conjugated with HRP.

In addition, the level of soluble B7-H3 (sB7-H3) in tumor cell (IMR5 and Panc-1) supernatant or mouse plasma was examined using Human B7-H3 Quantikine ELISA Kit (R&D Systems, DB7H30) according to the manufacturer's manual. Mouse plasma was harvested from 3 individual IMR5-bearing mice in each group of different B7-H3 CAR-T cells infusions.

## Antibody binding kinetics

The binding kinetics of anti-B7-H3 V_HH (B12/C4/G8)-Fc was performed on an Octet RED96 system (FortéBio) at 30 °C. 4IgB7-H3-his, human 2IgB7-H3-his, and mouse 2IgB7-H3-his proteins (R&D Systems) were respectively immobilized onto Ni-NTA sensor tips at 5 μg/ml for 120 s. The antigen-coated tips were then dipped into the buffer to stabilize the curve and subsequently dipped into 50 nM or 100 nM V_HH-hFc for association and dissociation measurements for a time window of 180 s and 300 s. Raw data was processed using Octet Data Analysis Software 9.0.

## Reverse transcriptase polymerase chain reaction (RT-PCR)

Total mRNA was isolated from multiple tumor cell lysates using a QuickPrep mRNA Purification kit (GE Healthcare), and first-strand cDNAs were synthesized using a SuperScript III First-Strand Synthesis System (Thermo Fisher Scientific) according to the manufacturer's instructions. Primers designed to amplify B7-H3 (4Ig and 2Ig isoforms) and β-actin were as follows (Supplementary Table 6). Human B7-H3-forward: 5′-GTGGTTCTGCCTCACAGGAG-3′; Human B7-H3-reverse:

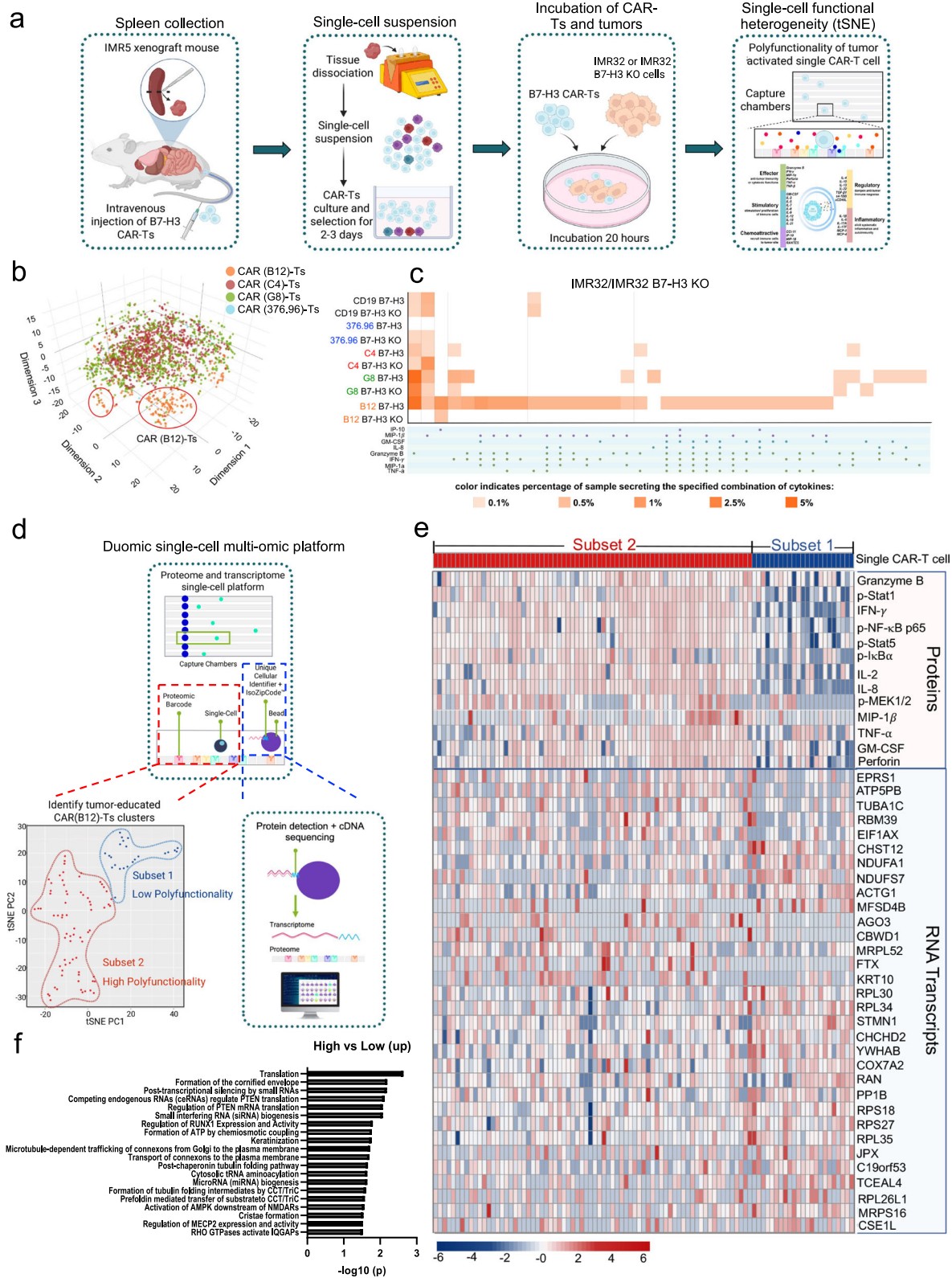

**Nature Communications** | (2023)14:5920

5′-ACCAGCAGTGCAATGAGACA-3′; Human β-actin-forward: 5′-CACCA ACTGGGACGACAT-3′; Human β-actin- reverse: 5′-ACAGCCTGGATA GCAACG-3′.

**Immunoblotting**

Protein was extracted from tumor cells using cell lysis buffer (Cell Signaling Technology), and protein concentration was measured using Bicinchoninic acid assay (Pierce) following the manufacturer's

manuals. PNGase F (New England BioLabs) was used to deglycosylate tumor protein samples following the manufacturer's protocol. 15 µg of cell lysates or 1 µg of recombinant B7-H3-his (4Ig and 2Ig) proteins (R&D Systems) were loaded onto 4–20% SDS-PAGE gel for electrophoresis. Three different commercial anti-B7-H3 antibodies and three anti-B7-H3 nanobodies were used: Anti-B7-H3 mAb (Abcam; #ab226256), anti-B7-H3 mAb (Cell Signaling Technology; #14058), 376.96 (Millipore Sigma), B12-Fc, C4-Fc, and G8-Fc. The anti-human

**Fig. 7 | Single-cell-based polyfunctional analysis of persistent CD4$^+$ B12(V$_H$H)-CAR-T cells by proteomics and transcriptomics expression. a** Flowchart of the persistent B7-H3 CAR-T cells polyfunctionality platform (created with BioRender.com). B7-H3 CAR-T cells were isolated from IMR5 xenograft mouse spleens followed by tumor cells (IMR32 or IMR32 B7-H3 KO) incubated for 20 hours ex vivo. These antigen-reactivated single-cell-based B7-H3 CAR-T cells were loaded into the 32-plex panel polyfunctionality capture platform, including cytokines/chemokines belonging to 5 groups: effector, stimulatory, chemoattractive, regulator, and inflammatory. **b** The t-distributed stochastic neighbor embedding (tSNE) plots demonstrated clustering obtained from 30,000 distinct long-persistent CD4$^+$ B7-H3 CAR-T cells from IMR5 xenograft mouse spleens using the 32-plex panel polyfunctionality capture platform. **c** Polyfunctional heat map displaying major functional cytokines/chemokines secreted across long-persistent CD4$^+$ B7-H3 CAR-T cells upon IMR32 and IMR32 B7-H3 KO stimulation from (**a**). **d** Flowchart of the Duomic single-cell multi-omic platform of long-persistent B12(V$_H$H)-CAR-T cells, which were identified into two clusters, subset 1-high polyfunctionality (72 single-cells) and subset 2-low polyfunctionality (23 single-cells). **e** The matched expression file of proteins and RNA transcripts in each single B12(V$_H$H)-CAR-T cell in either high or low polyfunctional B12(V$_H$H)-CAR-T subsets isolated from the Duomic platform. Transcriptomics expression profile of 32 candidate genes showed statistically significant expression differences ($P < 0.05$) between the high and low polyfunctional subsets. **f** REACTOME pathway analysis of unique genes in (**e**). The CAR-T cells used in this figure were produced using donor #076's PBMC.

GAPDH mAb was obtained from Cell Signaling Technology. The IgG-HRP-conjugated goat anti-human IgG mAb and IgG-HRP-conjugated goat anti-mouse IgG mAb (Jackson ImmunoResearch) were used as the secondary antibodies. For the epitope binding experiment, 1.5 μg of purified truncated B7-H3-his-flag fragments were respectively loaded onto 4–20% SDS-PAGE gel for electrophoresis and detected by anti-his-HRP mAb. V$_H$H-Fc or 376.96 were used to capture each fragment.

## Modeling of B7-H3 and nanobody complexes

To model the B7-H3 and nanobody complex structure, we predicted the nanobody structures using AbodyBuilder[49] webserver which also predicts the CDR residues to be used for docking. The B7-H3 structure was obtained from EMBL-EBI AlphaFold Protein Structure Database[50]. Docking was performed on HADDOCK2.4[51] where default parameters were used. The potential contact residues on B7-H3 were given according to the experimental peptide competition results. The complex with the highest HADDOCK score of each conformation was selected, and molecular graphics were performed in UCSF Chimera[52]. The scFv of 376.96 mAb was used in the modeling.

## Flow cytometry

Cell surface B7-H3 expression was detected by 376.96 mAb (Millipore Sigma) and a goat-anti-human IgG-APC (Jackson ImmunoResearch). To examine the binding ability of anti-B7-H3 antibodies to native cell surface B7-H3 in multiple tumor cells, we used 1 μg/ml V$_H$H-Fc (376.96 or MGA271 mAb) and detected with a goat-anti-human IgG-APC or a goat-anti-mouse IgG-APC (Jackson ImmunoResearch), respectively. The transduction efficiencies of B7-H3 CARs on T cells were detected by anti-EGFR human monoclonal antibody cetuximab (Erbitux) and goat-anti-human IgG-PE or APC-conjugated antibody. One hundred microlitres of blood were collected from mice and 1X RBC lysis buffer (eBiosciences) was used to remove red blood cells. To determine the absolute number of CAR-T cells in mouse blood, BV711 CD3, Erbitux, and goat-anti-human IgG conjugated with Alexa Fluor 488 (Biolegend) were used to stain CD3$^+$CAR$^+$-T cells. CAR-T cell count was measured using 123 count ebeads (ThermoFisher). For multiple-color staining, APC/Cy7-conjugated anti-CD4, FITC-conjugated anti-CD8, APC-conjugated anti-PD1, APC-conjugated anti-LAG3, APC-conjugated anti-CD62L, BV421-conjugated anti-TIM3, BV421-conjugated anti-CD45RA, PE-conjugated anti-CD95 and BV711 conjugated anti-CD3 (eBioscience) were used. The relative proportion of stem cell-like memory (Tscm: CD62L + CD45RA + CD95 +), central memory (Tcm: CD62L + CD45RA-CD95 +), effector memory (Tem: CD62L-CD45RA-CD95 +), and terminally differentiated effector memory (Temra: CD62L-CD45RA + CD95 +) subsets defined by CD62L, CD45RA, and CD95 expression in circulation CD4$^+$/CD8$^+$ CAR$^+$-T cell subpopulations. T cell exhaustion was evaluated via PE PD-1, PE TIM-3, and PE LAG-3 (Thermo Fisher Scientific). For the antigen-binding ability of B7-H3 CAR-T cells, 4IgB7-H3-Fc and 2IgB7-H3-Fc proteins at 10 μg/ml or 1 μg/ml were stained with different B7-H3 CAR-T cells or mock-Ts and was detected using a goat-anti-human IgG-APC. Data acquisition was performed using FACSCanto II (BD Biosciences) and SonySA3800 (Sony). Data were analyzed using FlowJo software (Tree Star).

## Generation of anti-B7-H3 CAR-T cells

We generated the B7-H3-targeted V$_H$H-based CAR and 376.96(scFv)-based CAR lentiviral vector following the design principle of the CAR construct published in our previous study[21]. Briefly, individual V$_H$H (B12/C4/G8) fragments or scFv of 376.96 mAb were respectively subcloned into a CAR construct (pMH338, pMH340, pMH339, and pMH372). The CAR-expressing lentivirus was produced as described previously[21]. PBMCs isolated from four healthy donors (#076, #1, #2, and #3) were stimulated for 24 hours using anti-CD3/anti-CD28 antibody-coated beads (Invitrogen) at a bead: cell ratio of 2:1 according to manufacturer's instructions in the presence of IL-2.

## In vitro cytolysis of B7-H3 CAR-T cells and activation assays

The cytotoxicity of B7-H3 CAR-T cells was determined by a luciferase-based assay. In brief, the luciferase-expressing tumor cells were used to establish a cytolytic assay. Cytolysis of B7-H3 CAR-T cells was detected by co-culturing with luciferase-expressing tumor cells (IMR5, IMR32, IMR32 B7-H3 KO, Panc-1, BxPC-3, NBEB, LAN-1, H226, and MDA-MB-231) at various E/T ratios for 24 hours followed by measurement of the luciferase activity using the luciferase assay system (Promega) on Victor (PerkinElmer). The supernatants were collected for TNF-α, IL-2, and IFN-γ detection using ELISA Kit (BioLegend). In the killing blocking assay of CAR-T cells, the varying concentration of soluble 4IgB7-H3 or 2IgB7-H3 protein was added into tumor CAR-T cells incubation for 48 h and 72 h.

## NF-κB/NFAT fluorescence reporter assay and confocal microscopy

The reporter gene (tdTomato) was engineered downstream NF-κB or NFAT site in the human Jurkat T cell line. Engineered NF-κB/NFAT-tdTomato Jurkat cells were transduced with different B7-H3 CARs. B12(V$_H$H)-CAR-Jurkat reporter cells were co-incubated with GFP overexpressed Panc-1 tumor cells at an E/T ratio of 1:1 in poly-L-lysine-coated μ-Slide for 5 hours. For live cell imaging, DRAQ5 reagent was used as a nuclear counterstain, and the slide was imaged using a 63× objective on a Zeiss LSM 880 Airyscan confocal microscope (Zeiss). For the X20 objective live cell imaging, we co-cultured different B7-H3 Jurkat reporter cells with Panc-1 cells for 24 hours at a 1:1 ratio. tdTomato expression level was measured through flow cytometry and quantified by Flowjo.

## Animal studies

5-week-old female NOD/SCID/IL-2Rgc$^{null}$ (NSG) mice were provided by the NCI CCR Animal Resource Program/NCI Biological Testing Branch, housed and treated under the protocol (LMB-059) approved by the Institutional Animal Care and Use Committee at the NIH. The Panc-1 i.v. model was established by i.v. injection of one million Panc-1 GL cells. Tumor-rechallenge was implemented in the tumor-free mice by i.v. injection of one million Panc-1 GL cells. The orthotopic pancreatic

tumor mouse models were established in the lab. A total of 0.25 million Panc-1 GL cells were directly injected into the tail of the mouse pancreas. The BxPC-3 i.p. model was established by i.p. injection of three million BxPC-3 GL cells, and the IMR5 i.v. model was established by i.v. injection of three million IMR5 GL cells. Mice were randomly allocated into several groups when the tumor signal was above $1 \times 10^9$ p/sec/cm$^2$/sr. The large NBEB s.c. tumor or NBEB B7-H3 KO s.c. tumor was implanted by s.c. injection of five million NBEB ML or NBEB B7-H3 KO cells in PBS mixed with Matrigel at a 1:1 ratio, respectively. Different doses (2.5 or 5 or 10 million) of B7-H3 CAR-T cells or irrelevant CD19 CAR-T cells were then i.v. injected once. Tumor volume was calculated as ½ (length × width$^2$) or bioluminescent intensity (Xenogen IVIS Lumina). Mice were euthanized by carbon dioxide ($CO_2$) inhalation method when they met any of the following endpoint criteria: tumor interfered with animals' ability to eat or drink, 20% weight loss, or any sign of outward distress such as hunched posture, ruffled fur, and reduced motility. A body condition scoring system out of 5 is used[53]. As well as a 4000 mm$^3$ tumor volume endpoint, a body condition score of <1.5/5, or a decreased activity level to the point of moribundity is used as a criterion for euthanasia.

### Single-cell cytokine profiling of T cell polyfunctionality

Ex vivo human T cells were isolated from IMR5-bearing mice spleens using Miltenyi Biotec tumor dissociation kit. Viable CAR-T cells were enriched using Ficoll. CD4$^+$/CD8$^+$ T cell subsets were separated using anti-CD4 or anti-CD8 microbeads (Miltenyi Biotec) and stimulated with B7-H3-positive IMR32 or IMR32 B7-H3 KO cells at a ratio of 1:2 for 20 h. Both CD4$^+$ and CD8$^+$ T populations were then recovered from the tumor cell-incubation systems. A single cell functional profile was determined as described previously[32,54]. Briefly, a total of 30,000 single cells were loaded onto an IsoCode chip (IsoPlexis) which contains ~12,000 microchambers. A 32-plex cytokine/chemokine antibody array was designed and embedded in each microchamber, including Effector: Granzyme B, IFN-γ, MIP-1α, Perforin, TNFα, TNFβ; Stimulatory: GM-CSF, IL-2, IL-5, IL-7, IL-8, IL-9, IL-12, IL-15, IL-21; Chemoattractive: CCL11, IP-10, MIP-1β, RANTES; Regulatory: IL-4, IL-10, IL-13, IL-22, sCD137, sCD40L, TGFβ1; Inflammatory: IL-1β, IL-6, IL-17A, IL-17F, MCP-1, MCP-4. The single-cell functional subsets within the different CD4$^+$ CAR-T cells groups were shown in a heatmap, and the combination of multiple cytokine/chemokine proteins in each sample was quantified as the percentage of polyfunctional cells.

### Single-cell proteomics and transcriptomics

Ex vivo IMR5 mice isolated CD4$^+$ B12(V$_H$H)-T cells were re-activated by incubating with IMR32 tumor cells at a E/T ratio of 1:2. We mainly focused on CD4$^+$ T subpopulation since we could only recover enough CD4$^+$ T cells from these IMR5 tumor-bearing mice. After a 24 h stimulation, a total of 3000 single-cell based CAR-T cells were loaded onto a Duomic chip (Isoplexis) with thousands of microchambers for single-cell proteomic evaluation. Various cytokines and phosphoproteins were captured and measured by fluorescence ELISA using a designed 18-plex antibody panel (including α-tubulin, GM-CSF, Granzyme B, INF-γ, IL-10, IL-2, IL-7, IL-8, MIP-1α, MIP-1β, p-IκBα, P-MEK1/2, p-NF-κB p65, p-Erk1/2, p-Stat1, p-Stat5, Perforin, and TNF-α) based on our CAR-T polyfunctionality data analysis, and which finally could separate CAR-T cells into high or low polyfunctional subsets. Simultaneously, the microchamber also houses an mRNA capture microbead with a unique molecular barcode sequence to capture mRNA molecules in the single-cell lysate. Reverse transcription occurs on the microbeads, and the synthesis double-stranded cDNA was concentrated/amplified by PCR, and then samples were sequenced on the Illumina NextSeq1000 instrument. Read lengths were as follows: 50 cycles read 1, 8 cycles index, 75 cycles read 2. Raw Fastq sequencing data was initially quality controlled with FASTQC version 0.11.9 to ensure reliable data quality. Cutadapt version 3.4 was utilized to trim nucleotide bases from read 1 that were irrelevant for cell barcode identification. The barcode sequence linked gene expression profile with corresponding protein detection from the same single cell in the microchamber. Subsequently, Alevin Salmon version 1.4.0 was utilized for cell barcode identification and gene expression quantification. Following Matching analysis, R version 4.0.3 was utilized to apply log normalization of matched IsoSpeak processed protein data. R package kmeans v.3.6.2 was subsequently utilized to perform. Kmeans clustering on data and Rtsne v. 0.15 were utilized to perform the T-SNE dimension reductional technique. Kmeans labels calculated on the protein data were then utilized to apply the same cell labels for the gene expression data. Differential Expression Analysis was performed utilizing DESeq2 v 1.30.1 with genes with a nominal p-value less than 0.05 considered to be significantly differentiated. Heatmaps were generated utilizing the R pheatmap library v. 1.0.12. Reactome Pathway analysis was performed utilizing Reactome Pathway Browser v 3.7.

### Histopathological and molecular pathology analysis

After CAR-T infusion (B7-H3 CAR-T cells or control CD19 CAR-T cells) on day 9, tumors were resected from the NBEB s.c. mice and fixed with 10% neutral buffered formalin and processed for routine hematoxylin and eosin (H&E) staining was conducted by Histoserv, Inc (Germantown, MD). Immunofluorescence and immunohistochemistry staining was performed by the Molecular Histopathology Laboratory (MHL) at the NCI. Briefly, after antigen retrieval with EDTA (Bond Epitope Retrieval 2) and endogenous peroxidase blocking, sections were incubated with anti-CD3 mAb (Bio-Rad, #MCA1477), followed by OPAL Fluorophore 520 (AKOYA). The anti-CD3 mAb complex was stripped by heating with Bond Epitope Retrieval 2. Sections were then incubated with anti-B7-H3 mAb (clone D9M2L, Cell Signaling Technology, #14058) followed by Bond Polymer reagent and OPAL Fluorophore 690 (AKOYA). Sections were removed from the Bond, DAPI stained, and coverslipped with Prolong Gold AntiFade Reagent (Invitrogen). Cleaved Caspase-3 (Cell Signaling Technology, #9661) immunohistochemistry was performed to predict cell apoptosis. Images were captured using the Aperio Scanscope FL (or XT) whole slide scanner, and staining was interpreted by a board-certified veterinary pathologist.

### Statistics

All experiments were repeated at least three times to ensure the reproducibility of results. All statistical analyses were performed using GraphPad Prism and are presented as mean±SEM. Results were analyzed using a 2-tailed unpaired Student's t-test. A P value of <0.05 was considered statistically significant. The number of repeats performed is described in the relevant figure legend.

### Reporting summary

Further information on research design is available in the Nature Portfolio Reporting Summary linked to this article.

## Data availability

Raw and processed data from the scRNA sequencing experiments are deposited and available in the NCBI's Gene Expression Omnibus (GEO) database under accession code GSE221411. The remaining data are available within the Article, Supplementary Information or Source Data files. Source data are provided with this paper.

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

## Acknowledgements

This work was supported by the Intramural Research Program of NIH, NCI, Center for Cancer Research (CCR) (Z01 BC010891 and ZIA BC010891 to M.H.) and the NCI CCR FLEX Program Synergy Award (to B.S.C., J.K., and M.H.). The dromedary camel $V_HH$ phage library construction was supported by the NIH DDIR Innovation Award (to M.H.). We thank the NCI CCR Animal Resource Program/NCI Biological ranch for providing the NSG mice used in the present study, NCI CCR/Leidos Animal Facility for assisting in animal support, the NHLBI Biophysics Core for assistance in antibody kinetics/affinity analysis, NCI CCR Confocal Microscopy Core for assistance in confocal microscopic analysis. NCI CCR Flow Cytometry Core Facility for assistance in cellular staining, and Frederick National Laboratory for Cancer Research Molecular Histopathology Laboratory for tissue staining service. We also thank the NIH Fellows Editorial Board for editorial assistance. The anti-B7-H3 camel $V_HH$ single domain antibodies presented in the present study are the subject of pending patent applications assigned to the NIH and are available for license in certain fields of use to qualified candidates. Please contact the corresponding author Dr. Mitchell Ho (NCI) at homi@nih.gov if you are interested in pursuing a license. This project has been funded in part with Federal funds from the National Cancer Institute, National Institutes of Health, under Contract No. HHSN261201800001I. The content of this publication does not necessarily reflect the views or policies of the Department of Health and Human Services, nor does mention of trade names, commercial products, or organizations imply endorsement by the U.S. Government.

## Author contributions

M.H. conceived the studies and supervised the project. D.L. and M.H. designed the studies and wrote the manuscript. D.L., R.W., T.L., and C.P. performed most of the experiments. H.R. made a phage-displayed camel VHH nanobody library. B.S.C provided the recombinant B7-H3 proteins of different species and knockout cell lines. E.E. analyzed the immunohistochemistry and immunofluorescence tissue staining. C.H.T. made structure models of the B7-H3/antibody complexes. W.N., J.Z., and S.M. conducted single-cell protein and RNA analysis of polyfunctional T cells. J.K., B.S.C., and M.H. provided funding and resources. All the authors reviewed, edited, and approved the manuscript.

## Competing interests

M.H., R.W., B.S.C., and D.L. are inventors on international patent application no. PCT/US2020/056601 (WO/2021/081052) assigned to the NIH, "High affinity nanobodies targeting B7-H3 (CD276) for treating multiple solid tumors". W.N., J.Z., and S.M. are employed by and have equity ownership in IsoPlexis. All authors declare no other competing interests.
