## [Peer Review File · Nature Communications]

Camel nanobody-based B7-H3 CAR-T cells show high efficacy against large solid tumoursREVIEWER COMMENTS

Reviewer #1 (Remarks to the Author):

The paper is very interesting and original in some aspects. The investigators demonstrated that the CAR-T cells based on the B12 VHH nanobody recognizing a unique epitope in the IgC domain of 4IgB7-H3 had the promising antitumor effect against advanced metastatic and large tumors in mice. They have used various tumor models to assess the potential benefit of nanobody-based B7-H3 CAR-T cells for treating solid tumors including pancreatic cancer and neuroblastoma, and provides a strategy for using nanobodies targeting rare and potent epitopes in engineering CAR-T cells.

Overall, the manuscript is of interest but there are some issues that need to be addressed or explained for a publication. The major issues are listed below:

1. The antigen-binding domains of CARs are generally single-chain variable fragments (scFv) targeting tumor-associated antigens, and used in most preclinical and clinical studies. It seems novel to use nanobody-based CAR-T to treat tumors. So, I wonder whether the nanobody-based CAR-T cells are safe for clinical situation? Since the tumor-killing potency of nanobody-based CAR-T cells are potent in this paper, do you have any plans to translate the findings to clinical trials?
2. What exactly is the "2Ig" of B7-H3?
3. One would like to see more representative cell models such as those from lung and breast cancers, in addition to pancreatic cancer and neuroblastoma? In the animal studies, there is only one B7-H3+ cell lines established in each model and there usually need more than one cell lines to verify one another, with two B7-H3 negative cell lines as control.
4. Jurkat is a malignant T cell line, but CAR-Jurkat is appropriate for evaluation of B7-H3 CAR activation that is proposed for standard CAR-T cells?
5. The percentage of Tcm and Tem subsets might be positively related to the persistent antitumor efficiency of CAR-T cells in vivo: any more rational or evidences for this conclusion?
6. Page 9, line 243: Increased apoptotic cells were observed in B12(VHH)-CAR-T rich areas. Please provide evidence to demonstrate that cells are apoptotic cells not necrosis cells?
7. Please explain the reason to isolate B7-H3 CAR-T cells from the spleens but not the tumors of IMR5 tumor-bearing mice at week 5 after CAR-T cell infusion?
8. Any explanation that the authors focused on mouse spleen-isolated CD4+CAR-T cells, rather than CD8+CAR-T cells inside the xenograft tumors?
9. Long-acting IMR5 tumor-educated B12(VHH)-CAR-T cells after 5-week treatment could be clustered into two groups with either high or low polyfunctionality versus single CAR-T cells): How? From where? CD4/CD8?
10. In figure 1e-g and 2b-f, how many times the experiment in each panel have been repeated.
11. Fig. 1g: The control should be culture media from B7-H3 negative cells?
12. In figure 5a and 6g, 10 million CAR-T cells are infused, in figure 5d and 6a, 5 million CAR-T cells are infused. Why the number of CAR-T cells infused are different? How to define the infusion amount?
13. In figure 6, the neuroblastoma (NBEB) xenograft mouse model, tumor volumes reached nearly 8000 mm³, is it in line with animal ethics?
14. In all mouse models, the toxicity of therapies should be analyzed.
15. In Extended Data Fig. 4, there should be a B7-H3 negative cell line to confirm the specificity.
16. Is the nanobody in the paper humanized and is necessary?

Reviewer #2 (Remarks to the Author):

Dan Li and colleagues from the NCI present a novel dromedary-based nanobody CAR T cell construct targeting a novel epitope within B7H3. B7H3 is a well-recognized tumor associated antigen with limited expression in normal tissues that has long been in the crosshairs of cancer

immunotherapeutics. Unfortunately, clinical responses to B7H3 targeted therapies have been limited.

The authors developed nanobody binders targeting the IgC, rather than the commonly targeted IgV, domain of B7H3 and show tumor responses in immunodeficient mice bearing pancreatic cancer and neuroblastoma both in metastatic and orthotopic models. No immune-competent models are provided.

In general, I found the manuscript to be extremely well written and logically organized. Figures were clear and readable.

I have only have one critique that would bear revision. The authors state in their metastatic and large tumor modeling that these experiments were done in order to mimic tumor microenvironments. However in the NSG models used, it is unclear how these mice would represent microenvironments seen in human pancreatic or neuroblastoma tumors. Furthermore, the authors then go on to use single cell transcriptomics and proteomics to hypothesize which genes may be involved in CAR T cell persistence.

I believe these findings, while interesting, need to be better contextualized given the models used. The conclusions should either be softened or immune competent models provided to support the claim that these experiments mimic tumor microenvironments.

Reviewer #3 (Remarks to the Author):

This is an interesting manuscript in which the authors describe the generation of a new antibody against B7-H3 for the generation of B7-H3 CAR T cells. They perform a comprehensive analysis *ex vivo* as well as in 2 animal models to select the most potent new B7-H3 antibody. The manuscript is well written, and the provided data support the conclusion of the authors. However, I have major concerns that primarily are focused on novelty.

Major:

1) Limited novelty: i) there are numerous studies that have reported on expression of B7-H3 in adult and pediatric solid tumors; ii) there are more than 20 manuscripts that have explored different designs of B7-H3-CARs in preclinical models; iii) the use of camelid antibodies to generate CAR T cells has been reported by numerous groups; iv) except for the antigen recognition domain, no novel CAR design is explored.

2) The authors state in the title that the generated CARTs are effective against large tumors; however the experiment in which large tumors are treated (6g) demonstrates that the novel CARTs do not induce regression of tumors, but show decrease in tumor growth. In addition, only 2 week of follow up is shown post CART infusion. This raises significant concern.

3) The studies shown in Figure 7 are impressive, but no mechanistic insights are gained.

Minor:

1) Figure 4: it is unclear if different donors were used for the functional analysis of CAR T cells.

2) Figure 7c: it is unclear how many times the analysis was performed.

3) Extended data Fig 4: only marker gene expression in CARTs is shown; please demonstrate CAR expression directly by Western blot or flow cytometry.

COMMENTS FROM THE REVIEWERS:

Reviewer #1: The paper is very interesting and original in some aspects. The investigators demonstrated that the CAR-T cells based on the B12 VHH nanobody recognizing a unique epitope in the IgC domain of 4IgB7-H3 had the promising antitumor effect against advanced metastatic and large tumors in mice. They have used various tumor models to assess the potential benefit of nanobody-based B7-H3 CAR-T cells for treating solid tumors including pancreatic cancer and neuroblastoma, and provides a strategy for using nanobodies targeting rare and potent epitopes in engineering CAR-T cells. Overall, the manuscript is of interest but there are some issues that need to be addressed or explained for a publication.

The major issues are listed below:

1. The antigen-binding domains of CARs are generally single-chain variable fragments (scFv) targeting tumor-associated antigens, and used in most preclinical and clinical studies. It seems novel to use nanobody-based CAR-T to treat tumors. So, I wonder whether the nanobody-based CAR-T cells are safe for clinical situation? Since the tumor-killing potency of nanobody-based CAR-T cells are potent in this paper, do you have any plans to translate the findings to clinical trials?

- We thank the reviewer's enthusiastic comments and suggestions.
- The low immunogenicity of nanobodies has been found in 20 patients with breast cancer in phase II clinical trials¹. So far, six CAR-T products have been approved by FDA for hematological cancers. Among them, the first five use the scFv as the antigen recognition domain, while the latest LCAR-B38M CAR-T (ciltacabtagene autoleucel) was derived from two camelid V_HHs (without humanization) targeting BCMA². LCAR-B38M CAR-T has been approved based on the finding of the open-label multicenter clinical trial CARTITUDE-1 (NCT03548207), in which the safety and efficacy of this CAR-T product were evaluated in 97 adult patients with relapsed/refractory multiple myeloma³. Camelid V_HH sequences are highly similar to those of human germline immunoglobulin V_Hs with nearly 80% identity; therefore, humanization of V_HHs (generally with less than a 5% increase of human residues) may not be essential before clinical translation. We included the immunogenicity of nanobodies in the Discussion section of the revised manuscript.
- In collaboration with the Pediatric Oncology Branch at the NIH Clinical Center, we are planning on the first-in-human clinical trial to utilize our B7-H3 (B12) nanobody CAR-T therapy for the treatment of pediatric cancers including neuroblastoma. The clinical trial is being scheduled to commence in 2024 and will be funded by the Cancer Moonshot program and the NCI Center for Cancer Research (CCR) Center for Cell-based Therapy.

Editorial Note: Top figure reprinted from Li, N., Spetz, M. R., Li, D. & Ho, M. *Advances in immunotherapeutic targets for childhood cancers: A focus on glypican-2 and B7-H3. Pharmacol Ther* **223**, 107892, doi:10.1016/j.pharmthera.2021.107892 (2021)., with permission from Elsevier.

2. What exactly is the “2Ig” of B7-H3?

- The 2Ig isoform has single extracellular V-like and C-like (VC) Ig domains, while 4Ig isoform consists of two sets of VC-like Ig domains. The sequences of the two IgV-IgC pairs are highly homologous (95% identity) due to exon duplication and differential splicing, making it difficult to isolate 2Ig-specific antibodies. We will show a diagram below to elucidate the “2Ig” and “4Ig” isoforms. This diagram is adapted from Figure 3A in our previous review article⁴. We cited this review article (Ref. 9) in the Introduction section of the revised manuscript.

3. One would like to see more representative cell models such as those from lung and breast cancers, in addition to pancreatic cancer and neuroblastoma? In the animal studies, there is only one B7-H3+ cell lines established in each model, and there usually need more than one cell lines to verify one another, with two B7-H3 negative cell lines as control.

- We thank the reviewer’s suggestions. We used two tumor cell lines representing lung cancer (H226) and breast cancer (MDA-MB-231) to examine the cytotoxicity of B7-H3 nanobody CAR-T cells. We added it in Supplementary Fig. 4B.

- In the animal study, we actually utilized three different tumor models: one pancreatic tumor model (Panc-1) (Fig. 5a) and two neuroblastoma models (IMR5 and NBEB) (Fig. 6a and 6g) to evaluate the anti-tumor efficacy of our B7-H3 nanobody CAR-T cells. Additionally, in the revised manuscript we established a new pancreatic cancer mouse model using the BxPC-3 cell line and allowed tumors to grow to a large size (with an average tumor signal of 5×10^{10} p/sec/cm²/sr, equivalent to around 250 mm³ tumors) before CAR-T treatment. As demonstrated below, the B12(V_HH)-CAR-T cells exhibited the best antitumor efficacy, followed by C4(V_HH)-CAR-T cells, while 376.96(V_HH)-CAR-T cells had limited efficacy. We added this new exciting data in Fig. 6k-m.

- We agree with the reviewer's suggestions. To assess the antigen-specificity of B7-H3 CAR-T cells, we performed *in vitro* validation by incubating them with IMR32 B7-H3 knockout (KO) cells (Fig. 4b). However, establishing a mouse xenograft model using this KO line was challenging due to poor tumorigenicity and very slow growth of IMR32 B7-H3 KO cells in mice. Therefore, we decided to establish a mouse xenograft model by subcutaneous injection of NBEB B7-H3 KO cells for the revised manuscript. Our data showed that 10 million B12(V_HH)-CAR-T cells could inhibit NBEB tumor growth (Fig. 6h) but not NBEB B7-H3 KO tumor, indicating that the anti-tumor activity of our nanobody-based B7-H3 CAR-T cells is antigen-dependent. We added this data in Supplementary Fig. 8 and described it in the Results section.

4. Jurkat is a malignant T cell line, but CAR-Jurkat is appropriate for evaluation of B7-H3 CAR activation that is proposed for standard CAR-T cells?

- Activation and nuclear translocation of transcription factors are significant downstream consequences of TCR signaling. To evaluate the activation of various B7-H3 nanobody-based CAR, we established the NF- κ B or NFAT reporter assay using the Jurkat line. Jurkat is widely utilized as a T-cell model to investigate T cell activation and signaling mechanisms⁵. Jurkat-based reporter systems have been employed to examine the effects of external stimuli on the activation of transcription factors such as NF- κ B, NFAT and AP-1⁶. Although Jurkat is not a primary T-cell line and could have potential constitutive activation of PI3K-mediated signal-transduction pathways⁷, it's still a cost-efficient and rapid method for evaluating the signaling of CARs in human T cells⁸. Using this approach, we observed that B12(V_HH)-CAR-Jurkat exhibited the highest potency in T-cell activation when stimulated with tumor cells, which correlated with its cytotoxic ability.

5. The percentage of T_{cm} and T_{em} subsets might be positively related to the persistent antitumor efficiency of CAR-T cells in vivo: any more rational or evidences for this conclusion?

- We thank the reviewer's question. In our study, we treated Panc-1 mice with B7-H3 CAR-T cells and found a high percentage of T_{cm} and T_{em} proportions in the persistent nanobody-based CD8⁺ CAR-T cells circulated in the mouse blood at week 6 after infusion. Moreover, we also found that the circulating CD8⁺ B12(V_HH)-CAR-T cells comprised a higher percentage of the T_{em} subpopulation than other treatment groups but G8(V_HH)-

CAR-T cells showed a lower percentage of Tcm. Therefore, we concluded that the Tem and Tcm subpopulations might be crucial to the high-efficient antitumor activity of persistent CAR-T cells.

- In general, the memory T cells include stem cell memory T (Tscm) cells, central memory T (Tcm) cells, and effector memory T (Tem) cells, which have different specific phenotypes. The efficacy of adoptive cell-transfer therapy requires that transferred T cells persist *in vivo*. Tem cells express higher levels of receptors responsible for migration to inflamed tissues and have a stronger immediate effector function than other cell types, which can explain why we observed a higher percentage of the Tem subpopulation in the most effective CD8⁺ B12(V_HH)-CAR-T cells than other treatment groups. Tcm cells persisted longer and showed a superior antitumor activity than Teff cells in mice^{9,10}. Sommermeyer et al. isolated memory T cells from B cell malignancies and generated CD19 CAR-T cells. They demonstrated that the Tcm-derived CD8⁺ CAR T cells have superior survival and greater proliferation capacity *in vivo* than CD8⁺ Tn (naïve T cells)- and CD8⁺ Tem-derived CAR T cells¹¹. Therefore, we concluded that the Tem and Tcm subpopulations might be crucial to the high-efficient antitumor activity of persistent CAR-T cells. We discussed it in the Discussion section of the revised manuscript.

6. Page 9, line 243: Increased apoptotic cells were observed in B12(V_HH)-CAR-T rich areas. Please provide evidence to demonstrate that cells are apoptotic cells not necrosis cells?

- We thank the reviewer's suggestion. To ensure accurate measurement of apoptotic cells, we worked with a pathologist (E. Edmondson) from the NCI Molecular Histopathology Core. We did immunohistochemistry (IHC) by using Caspase-3 as a cell apoptosis marker, and we found increased apoptotic cells in the densely infiltrated B12(V_HH)-CAR-T areas. The apoptosis mechanism is favored based on H&E morphologic features, including cell shrinkage, nuclear shrinkage/condensation, nuclear pyknosis with apoptotic bodies, etc. In the revised manuscript, we showed the colocalization of increased apoptotic cells (arrow) stained by Caspase-3 (IHC) and infiltrated B12(V_HH)-CAR-T cells in the NBEB tumor tissues in two distinct areas. We added this data in Supplementary Fig. 10.

7. Please explain the reason to isolate B7-H3 CAR-T cells from the spleens but not the tumors of IMR5 tumor-bearing mice at week 5 after CAR-T cell infusion?

- We thank the reviewer's question. CAR-T cell functions and persistence are critical factors influencing their antitumor immune response in clinical settings. Due to the limited number of persistent CAR-T cells from tumors after week 5 of infusion, we opted to utilize spleens for our analysis. Moreover, the small size of the tumors hindered their collection or dissociation from the B12 CAR-T-treated mice at this stage. Notably, recent clinical trials have shown that CAR-T cells predominantly homed to the lungs and spleen¹², and preclinical studies have demonstrated that the spleen is the site of their most significant proliferation compared to other organs such as blood, lung, kidney, liver, heart, and bone marrow¹³. Therefore, we isolated CAR-T cells from the mouse spleen for examination.

8. Any explanation that the authors focused on mouse spleen-isolated CD4+CAR-T cells, rather than CD8+CAR-T cells inside the xenograft tumors?

- We thank the reviewer's important question. In our study, we found that the CAR-T population isolated from IMR5 mouse spleen at week 5 of infusion consisted of mainly CD4+ CAR-T cells (84%±3.0) than CD8+ CAR-T subset (11%±1.0). This prompted us to focus primarily on CD4+ CAR-T subset. We performed the analysis on the recovered CD8+ CAR-T cells (see figure below), but due to limited cell numbers, a substantial population of polyfunctional CD8+ CAR-T cells was not observed. A comparable CD4+/CD8+ ratio was also observed in healthy donor-derived PMBCs used for B7-H3 CAR-T cell production. We added this data in Supplementary Fig. 12.

color indicates percentage of sample secreting the specified combination of cytokines:

- To understand the long-term potential and clonal stability of the infused cells, Melenhorst JJ and his colleagues studied long-lasting CD19 CAR-T cells in two patients with chronic lymphocytic leukemia who achieved complete remission in 2010 and published their work in *Nature* in 2022¹⁴. They found CAR T cells remained detectable more than ten years after infusion and interestingly, CD4⁺ CAR T cell population was dominant at the later time points. They also demonstrated that CD4⁺ CAR T cells exhibited cytotoxic characteristics along with ongoing functional activation and proliferation. This work may explain why there are more CD4⁺ than CD8⁺ in the long-persistent CAR-T cells. We added the new information in the Discussion section of the revised manuscript. We plan to further evaluate the biological significance of the CD4⁺ T cells within the tumor microenvironment in future clinical investigations.

9. Long-acting IMR5 tumor-educated B12(V_HH)-CAR-T cells after 5-week treatment could be clustered into two groups with either high or low polyfunctionality versus single CAR-T cells): How? From where? CD4/CD8?

- In this study, we mainly focused on CD4⁺ CAR T subpopulation rather than CD8⁺ since we could only recover sufficient long-persistent CD4⁺ CAR T cells from IMR5 tumor-bearing mice. We explained it in the Methods (Single-cell proteomics and transcriptomics) section. We also described how to separate B12(V_HH)-CAR-T cells into two subsets based on the T cell polyfunctionality, in the Methods (Single-cell proteomics and transcriptomics) section. Briefly, these IMR5 mouse-isolated CD4⁺ B12(V_HH)-CAR-T cells were re-

activated by incubating with IMR32 tumor cells *in vitro*. A total of 3,000 single CAR-T cells were loaded onto a Duomic chip (Isoplexis) with thousands of microchambers for single-cell proteomic evaluation. Various cytokines and phosphoproteins were captured and measured by fluorescence ELISA using a designed 18-plex antibody panel based on our CAR-T polyfunctionality data, which finally could separate CAR-T cells into high or low polyfunctional subsets for single-cell RNA-based transcriptome analysis.

10. In figure 1e-g and 2b-f, how many times the experiment in each panel have been repeated.

- We did at least three independent repeats for these experiments. We added the number of repeats in each panel in the figure legend of the revised manuscript. For the data of B7-H3 mRNA expression (Fig. 1e), B7-H3 protein expression (Fig. 1f), B7-H3/V_HH-Fc interaction (Fig. 2d), and antigen-binding ability of V_HH-Fc (Fig. 2f), we did more repeats to optimize the specificity of primers and concentrations of anti-B7-H3 antibodies. For the ELISA assay in Fig. 1g, 2b, and 2e, we performed three independent experiments for statistical analyses. We added this information to the figure legends.

11. Fig. 1g: The control should be culture media from B7-H3 negative cells?

- Besides native culture media (CM), we incorporated an additional negative control comprising the supernatant collected from a time course of *in vitro* cultured IMR32 B7-H3 knockout (KO) cells. Based on the data shown below, we demonstrated a significantly increased level of sB7-H3 protein in the cell culture systems of Panc-1, IMR5, and IMR32 over time compared with culture media and the culture supernatant of IMR32 B7-H3 KO cells. We updated Fig. 1g in the revised manuscript.

12. In figure 5a and 6g, 10 million CAR-T cells are infused, in figure 5d and 6a, 5 million CAR-T cells are infused. Why the number of CAR-T cells infused are different? How to define the infusion amount?

- We determined the CAR-T cell dose for each tumor model based on disease burden and the data from pilot mouse testing that may mitigate toxicity while maintaining efficacy¹⁵.
- To optimize the CAR-T treatment dose, multiple experiments were conducted in Panc-1 mouse models. Initially, we treated Panc-1 mice with 10 million CAR-T cells (Fig. 5a), followed by gradually decreasing the dose to 5 million (Fig. 5d) and 2.5 million (Fig. 5i) cells. Based on the findings, 5 million CAR-T cells were identified as the standard dose for B7-H3 CAR-T treatment. This dose was subsequently used in the IMR5 mouse model (Fig. 6a). Additionally, to explore the potential of B7-H3 CAR-T cells in treating large tumors, we established subcutaneous NBEB tumors known for their aggressive growth in mice, allowing us to infuse 10 million CAR-T cells (Fig. 6g).

13. In figure 6, the neuroblastoma (NBEB) xenograft mouse model, tumor volumes reached nearly 8000 mm³, is it in line with animal ethics?

- We thank the reviewer's important question. According to our animal protocol approved by the Institutional Animal Care and Use Committee at NIH, mice were euthanized when they met any of the following endpoint criteria: tumor interfered with animals' ability to eat or drink, 20% weight loss, or any sign of outward distress such as hunched posture, ruffled fur, and reduced motility. In the clinic, cancer patients usually grow very large

tumors that are difficult to treat using current FDA approved drugs. Since in this study we found our B7-H3 nanobody CAR-T cells were very active, we decided to test them on large tumors. The mice bearing large tumors are monitored twice daily by the laboratory, including on weekends and holidays. These cages are identified by the investigator by using a cage flag on the cage and twice daily monitoring of animals' health is recorded on a log in the animal housing room. Skin ulceration and tumor necrosis are expected as a result of treatment or tumor progression, and animals are monitored for body condition and activity level. Animals that have ulceration or necrosis should have additional documentation of their condition on the Ulcerated Tumor Checklist, also in the animal housing room. Tumor skin ulceration or necrosis that progresses beyond the margins of the tumor (i.e., into normal tissue), is a criterion for euthanasia. Body weight is also monitored. However, tumor shrinkage can result in significant body weight loss, therefore, body condition scoring is a more appropriate measure in terms of animal condition. A body condition scoring system out of 5 is used¹⁶. Training in the appropriate implementation of this scoring system is done by veterinary technical staff. As well as a 4000 mm³ tumor volume endpoint, a body condition score of < 1.5/5, or a decreased activity level to the point of moribundity is a criterion for euthanasia. We incorporated the endpoint statement in the Methods (Animal studies) section.

14. In all mouse models, the toxicity of therapies should be analyzed.

- Throughout our study, the mice treated with B7-H3 CAR-T cells remained healthy until the completion of a mouse experiment or sacrifice. However, we occasionally observed cases of graft-versus-host disease (GVHD) in mice after week 3 of CAR-T infusion, particularly in the first mouse experiment involving Panc-1 tumor-bearing mice treated with 10 million CAR-T cells. Following our animal protocol, these mice were euthanized.

15. In Extended Data Fig. 4, there should be a B7-H3 negative cell line to confirm the specificity.

- We confirmed the killing specificity of B7-H3 CAR-T cells by conducting *in vitro* experiments using both IMR32 cells and IMR32 B7-H3 KO cells (Fig. 4b). The results demonstrated that the cytotoxicity of anti-B7-H3 V_HH-based CAR-T cells is antigen-

dependent. Furthermore, to validate the antigen specificity of B12(V_HH)-CAR-T, our most promising candidate, we established a subcutaneous NBEB B7-H3 KO mouse model and conducted *in vivo* experiments (Supplementary Fig. 8 in the revised manuscript).

16. Is the nanobody in the paper humanized and is necessary?

- We thank the reviewer's great question. The camel-derived nanobodies generated in this study are not humanized. Humanization of mouse or rabbit antibodies is a common approach to reduce immunogenicity risk in humans as we previously did^{17,18}. However, the impact of nanobody V_HH humanization on immunogenicity remains uncertain, as some V_HHs show no immunogenicity while certain fully human VHs can elicit anti-drug antibodies¹⁹. LCAR-B38M (*ciltacabtagene autoleucel*), the recently FDA-approved CAR-T product for myeloma treatment, utilizes bivalent camelid V_HHs derived from llama targeting BCMA without undergoing humanization. Based on this, we consider humanization to be advantageous for nanobodies entering clinical trials, but we also recognize that it may not always be essential. Camelid V_HH sequences are highly similar to those of human immunoglobulin germline VHs with nearly 80% identity; therefore, humanization of V_HHs (generally with less than 5% increase of human residues) may not be necessary before clinical translation. We added the immunogenicity information in the Discussion section of the revised manuscript.

Reviewer #2: Dan Li and colleagues from the NCI present a novel dromedary-based nanobody CAR T cell construct targeting a novel epitope within B7H3. B7H3 is a well-recognized tumor associated antigen with limited expression in normal tissues that has long been in the crosshairs of cancer immunotherapeutics. Unfortunately, clinical responses to B7H3 targeted therapies have been limited.

The authors developed nanobody binders targeting the IgC, rather than the commonly targeted IgV, domain of B7H3 and show tumor responses in immunodeficient mice bearing pancreatic cancer and neuroblastoma both in metastatic and orthotopic models. No immune-competent models are provided.

In general, I found the manuscript to be extremely well written and logically organized. Figures were clear and readable.

I have only have one critique that would bear revision. The authors state in their metastatic and large tumor modeling that these experiments were done in order to mimic tumor microenvironments. However in the NSG models used, it is unclear how these mice would

represent microenvironments seen in human pancreatic or neuroblastoma tumors. Furthermore, the authors then go on to use single cell transcriptomics and proteomics to hypothesize which genes may be involved in CAR T cell persistence.

I believe these findings, while interesting, need to be better contextualized given the models used. The conclusions should either be softened or immune competent models provided to support the claim that these experiments mimic tumor microenvironments.

- We appreciate enthusiastic comments from the reviewer. We revised the conclusion by removing the mention of “tumor microenvironment” in the revised manuscript.
- We described the rationale of the different mouse xenografts in this study. The majority of PDAC patients have metastatic disease at the time of diagnosis. To mimic metastatic PDAC, we used Panc-1, a commonly used pancreatic cancer cell line, to establish a mouse xenograft via intravenous injection²⁰. Metastatic relapse is the major cause of death in neuroblastoma (NB). Hence, we established a metastatic IMR5 mouse model as described previously that shows a similar pattern of experimental metastasis as occurs in NB patients²¹. We also established mouse models bearing large NB tumors by subcutaneous injection of NBEB tumor cells to explore the infiltration of nanobody CAR-T cells. Additionally, in the revised manuscript we established a new pancreatic cancer mouse model using the BxPC-3 cell line and allowed tumors to grow to a large size (with an average tumor signal of 5×10^{10} p/sec/cm²/sr, equivalent to the tumor size of around 250 mm³) before CAR-T treatment. All the mice we used were immunodeficient (NSG) mice.
- In this study, we demonstrated that the B7-H3 is a promising target for CAR-T therapy in treating multiple solid tumors. In addition to being highly expressed in tumor cells, B7-H3 was also abundant in stromal cells and tumor vasculature, inhibiting lymphocytic infiltration in the tumor microenvironment (TME)^{22,23}. To better understand the role of B7-H3 in cancer therapy and enhance its translational impact, future research should evaluate the potency and mechanism of B7-H3 CAR-T cells in the TME using immunocompetent mouse tumor models, such as KPC mouse model for pancreatic cancer²⁴ and TH-MYCN transgenic murine model for neuroblastoma²⁵. We added this in the Discussion section of the revised manuscript.

Reviewer #3: This is an interesting manuscript in which the authors describe the generation of a new antibody against B7-H3 for the generation of B7-H3 CAR T cells. They perform a

comprehensive analysis ex vivo as well as in 2 animal models to select the most potent new B7-H3 antibody. The manuscript is well written, and the provided data support the conclusion of the authors. However, I have major concerns that primarily are focused on novelty.

Major:

1) Limited novelty: i) there are numerous studies that have reported on expression of B7-H3 in adult and pediatric solid tumors; ii) there are more than 20 manuscripts that have explored different designs of B7-H3-CARs in preclinical models; iii) the use of camelid antibodies to generate CAR T cells has been reported by numerous groups; iv) except for the antigen recognition domain, no novel CAR design is explored.

- We thank the reviewer's enthusiastic comments. We would like to highlight the novelty of our study here and in the first paragraph of the Discussion section in the revised manuscript.
 - i) We agree with the reviewer that the antigen recognition domain is novel in our CAR-T design. One of the major obstacles in CAR generation is the optimization of antigen recognition domains. The location of an antibody binding site on the target is critical for the efficacy of CARs. However, it is generally unknown what epitope works best for a cancer target. Our anti-B7-H3 nanobodies were isolated from eight large dromedary camel phage libraries constructed in our laboratory, not from llama or alpaca as most other groups do. Our sequencing data indicate that camels generally have more diverse V_HH sequences than those from llamas and alpacas due to a large camel V_HH repertoire. In this study, using large camel V_HH libraries to isolate nanobodies that recognize all the epitope domains on a tumor antigen, we identified a new epitope that enabled the development of nanobody-based CAR-T cells with activity against large tumors. Our B7-H3 nanobody CAR-T cells derived from the camel B12 V_HH reported in this manuscript are very active as compared to other CAR-T cells in large solid tumor models. We compared our anti-B7-H3 V_HH-based CAR-T with one of the most well-known 376.96(scFv)-CAR-T targeting B7-H3. Our results showed that V_HH-based CAR-T cells, targeting the IgC domain rather than the IgV domain, exhibited greater potency in preclinical tumor models, including two large tumor models (NBEB and BxPC-3) for neuroblastoma and pancreatic cancer. The tumors in this size are normally not sensitive to most targeted therapies, including CAR-T therapy. To our knowledge,

there are currently no published nanobody-based CAR-T cells targeting B7-H3 for cancer immunotherapy.

- ii) We confirmed that the dominant isoform of B7-H3 in cancer is 4Ig based on RNA and protein expression analysis. Our experiments using B7-H3⁺ and B7-H3⁻ tumor cell lines showed no protein expression of the 2Ig isoform, contrary to the previous study²⁶. Additionally, we demonstrated that the anti-B7-H3 antibody used by another research group to probe the expression of 2Ig isoform in normal brain tissues was not specific for B7-H3. Our data suggests that only the 4Ig isoform should be the target of cancer immunotherapy. This validation is important for understanding biology of B7-H3 and developing effective cancer therapy.
- iii) To analyze the functionality of CAR-T cells, we used a new cutting-edge assay based on single-cell transcriptome RNA sequencing coupled with single-cell functional proteomics and identified a panel of upregulated genes in the polyfunctional CAR-T cells harvested from mice. We also made a reporter assay to visualize and analyze the function of CAR-T cells for NF- κ B and NFAT signaling. These new assays help us understand the regulation of CAR-T cells by transcriptional, translational, and T-cell signaling responses to the tumors in an antigen-specific manner.

2) The authors state in the title that the generated CARTs are effective against large tumors; however the experiment in which large tumors are treated (6g) demonstrates that the novel CARTs do not induce regression of tumors, but show decrease in tumor growth. In addition, only 2 week of follow up is shown post CART infusion. This raises significant concern.

- We thank the reviewer's comments. It is challenging to monitor the growth of a large tumor for an extended duration. According to our animal protocol approved by the Institutional Animal Care and Use Committee at NIH, mice were euthanized when they met any of the following endpoint criteria: tumor interfered with animals' ability to eat or drink, 20% weight loss, or any sign of outward distress such as hunched posture, ruffled fur, and reduced motility. Since the tumor grew very rapidly (doubling the size every other day), we had to obtain special approval and monitor the mice bearing large tumors twice daily including on weekends and holidays. A body condition scoring system out of 5 is used¹⁶. As well as a 4000 mm³ tumor volume endpoint, a body condition score of < 1.5/5, or a

decreased activity level to the point of moribundity is a criterion for euthanasia. We incorporated this endpoint statement in the Methods (Animal studies) section. Therefore, we could not monitor the mice for a longer duration due to the fact that they already reached one of the endpoint criteria listed in our animal protocol.

- In the revised manuscript, we established a new pancreatic tumor model by i.p. injection of BxPC-3 tumor cells and treated the mice at a high tumor signal of approximately 5×10^{10} . Based on our previous experience with this model, the starting tumor size in this experiment was around 250 mm³. Our results showed that 5 million B12(V_HH)-CAR-T cells effectively reduced tumor burden, while 376.96(scFv)-CAR-T showed limited efficacy. C4(V_HH)-CAR-T cells inhibited tumor growth in 3 out of 5 mice. The new exciting results have been included in Fig. 6k-m.

3) The studies shown in Figure 7 are impressive, but no mechanistic insights are gained.

- We appreciate the reviewer's enthusiastic comments on our single-cell-based analysis of persistent CAR-T. Our findings based on single-cell proteomics and transcriptomics expression analysis revealed several key factors associated with the high-polyfunctional persistent B12(V_HH)-CAR-T cells. Notably, the expression of cytokines including granzyme B, perforin, IFN- γ , IL-2, TNF- α was consistently elevated in high-polyfunctional B12(V_HH)-CAR-T cells as compared to low-polyfunctional cells. Additionally, we observed increased phosphorylation of NF- κ B p65 in the high-polyfunctional B12(V_HH)-CAR-T cells, which is consistent with our observation in the CAR-Jurkat NF- κ B reporter assay, suggesting the effective activation of NF- κ B signaling in high-polyfunctional B12(V_HH)-CAR T cells.

- At the RNA level, we listed top genes that are significantly up-regulated in the high-polyfunctional population as compared to the low-polyfunctional population and explained their functions in the Discussion section of the revised manuscript. One notable transcript is *EPRSI*, which plays a crucial role in protein synthesis and metabolism, particularly in response to IFN- γ stimulation. T-cell activation and differentiation are accompanied by dramatic changes in cellular metabolic programs. These changes may represent a mechanism of “metabolic checkpoint” that coordinates metabolic status with cellular signaling in the tumor microenvironment²⁷. In addition to *EPRSI*, other up-regulated transcripts in the high-polyfunctional CAR-T, such as *MRPL52*, *EIF1AX*, *ATP5PB*, *TUBA1C*, *CBWD1*, and *MFSD4B*, are related to T-cell metabolic system. The new information has been incorporated into the Discussion section of the revised manuscript.

Minor:

1) Figure 4: it is unclear if different donors were used for the functional analysis of CAR T cells.

- We used the same donor for the whole manuscript and clarified it in the figure legends.

2) Figure 7c: it is unclear how many times the analysis was performed.

- We used 3,000 single cells to do the functional analysis. We described it in the Methods section (Single-cell proteomics and transcriptomics).

3) Extended data Fig 4: only marker gene expression in CARTs is shown; please demonstrate CAR expression directly by Western blot or flow cytometry.

- We measured CAR expression directly by flow cytometry using antigen protein as shown in Fig. 4e.

References

- 1 Ackaert, C. *et al.* Immunogenicity Risk Profile of Nanobodies. *Front Immunol* **12**, 632687, doi:10.3389/fimmu.2021.632687 (2021).
- 2 Mullard, A. FDA approves second BCMA-targeted CAR-T cell therapy. *Nat Rev Drug Discov* **21**, 249, doi:10.1038/d41573-022-00048-8 (2022).
- 3 Berdeja, J. G. *et al.* Ciltacabtagene autoleucel, a B-cell maturation antigen-directed chimeric antigen receptor T-cell therapy in patients with relapsed or refractory multiple

- myeloma (CARTITUDE-1): a phase 1b/2 open-label study. *Lancet (London, England)* **398**, 314-324, doi:10.1016/S0140-6736(21)00933-8 (2021).
- 4 Li, N., Spetz, M. R., Li, D. & Ho, M. Advances in immunotherapeutic targets for childhood cancers: A focus on glypican-2 and B7-H3. *Pharmacol Ther* **223**, 107892, doi:10.1016/j.pharmthera.2021.107892 (2021).
- 5 Abraham, R. T. & Weiss, A. Jurkat T cells and development of the T-cell receptor signalling paradigm. *Nat Rev Immunol* **4**, 301-308, doi:10.1038/nri1330 (2004).
- 6 Jutz, S. *et al.* Assessment of costimulation and coinhibition in a triple parameter T cell reporter line: Simultaneous measurement of NF-kappaB, NFAT and AP-1. *J Immunol Methods* **430**, 10-20, doi:10.1016/j.jim.2016.01.007 (2016).
- 7 Astoul, E., Edmunds, C., Cantrell, D. A. & Ward, S. G. PI 3-K and T-cell activation: limitations of T-leukemic cell lines as signaling models. *Trends Immunol* **22**, 490-496, doi:10.1016/s1471-4906(01)01973-1 (2001).
- 8 Bloemberg, D. *et al.* A High-Throughput Method for Characterizing Novel Chimeric Antigen Receptors in Jurkat Cells. *Mol Ther Methods Clin Dev* **16**, 238-254, doi:10.1016/j.omtm.2020.01.012 (2020).
- 9 Berger, C. *et al.* Adoptive transfer of effector CD8+ T cells derived from central memory cells establishes persistent T cell memory in primates. *The Journal of clinical investigation* **118**, 294-305, doi:10.1172/JCI32103 (2008).
- 10 Klebanoff, C. A., Gattinoni, L. & Restifo, N. P. CD8+ T-cell memory in tumor immunology and immunotherapy. *Immunol Rev* **211**, 214-224, doi:10.1111/j.0105-2896.2006.00391.x (2006).
- 11 Sommermeyer, D. *et al.* Chimeric antigen receptor-modified T cells derived from defined CD8+ and CD4+ subsets confer superior antitumor reactivity in vivo. *Leukemia* **30**, 492-500, doi:10.1038/leu.2015.247 (2016).
- 12 Ritchie, D. S. *et al.* Persistence and efficacy of second generation CAR T cell against the LeY antigen in acute myeloid leukemia. *Mol Ther* **21**, 2122-2129, doi:10.1038/mt.2013.154 (2013).
- 13 Ying, Z. *et al.* Distribution of chimeric antigen receptor-modified T cells against CD19 in B-cell malignancies. *BMC cancer* **21**, 198, doi:10.1186/s12885-021-07934-1 (2021).
- 14 Melenhorst, J. J. *et al.* Decade-long leukaemia remissions with persistence of CD4(+) CAR T cells. *Nature* **602**, 503-509, doi:10.1038/s41586-021-04390-6 (2022).
- 15 Dasyam, N., George, P. & Weinkove, R. Chimeric antigen receptor T-cell therapies: Optimising the dose. *Br J Clin Pharmacol* **86**, 1678-1689, doi:10.1111/bcp.14281 (2020).
- 16 Ullman-Cullere, M. H. & Foltz, C. J. Body condition scoring: a rapid and accurate method for assessing health status in mice. *Lab Anim Sci* **49**, 319-323 (1999).
- 17 Zhang, Y. F. & Ho, M. Humanization of rabbit monoclonal antibodies via grafting combined Kabat/IMGT/Paratome complementarity-determining regions: Rationale and examples. *mAbs* **9**, 419-429, doi:10.1080/19420862.2017.1289302 (2017).
- 18 Zhang, Y. F. & Ho, M. Humanization of high-affinity antibodies targeting glypican-3 in hepatocellular carcinoma. *Scientific reports* **6**, 33878, doi:10.1038/srep33878 (2016).
- 19 Rossotti, M. A., Belanger, K., Henry, K. A. & Tanha, J. Immunogenicity and humanization of single-domain antibodies. *FEBS J* **289**, 4304-4327, doi:10.1111/febs.15809 (2022).
- 20 Miquel, M., Zhang, S. & Pilarsky, C. Pre-clinical Models of Metastasis in Pancreatic Cancer. *Front Cell Dev Biol* **9**, 748631, doi:10.3389/fcell.2021.748631 (2021).

- 21 Li, N., Nguyen, R., Thiele, C. J. & Ho, M. Preclinical testing of chimeric antigen receptor T cells in neuroblastoma mouse models. *STAR Protoc* **2**, 100942, doi:10.1016/j.xpro.2021.100942 (2021).
- 22 MacGregor, H. L. *et al.* High expression of B7-H3 on stromal cells defines tumor and stromal compartments in epithelial ovarian cancer and is associated with limited immune activation. *J Immunother Cancer* **7**, 357, doi:10.1186/s40425-019-0816-5 (2019).
- 23 Seaman, S. *et al.* Eradication of Tumors through Simultaneous Ablation of CD276/B7-H3-Positive Tumor Cells and Tumor Vasculature. *Cancer Cell* **31**, 501-515 e508, doi:10.1016/j.ccell.2017.03.005 (2017).
- 24 Lee, J. W., Komar, C. A., Bengsch, F., Graham, K. & Beatty, G. L. Genetically Engineered Mouse Models of Pancreatic Cancer: The KPC Model (LSL-Kras(G12D/+);LSL-Trp53(R172H/+);Pdx-1-Cre), Its Variants, and Their Application in Immuno-oncology Drug Discovery. *Curr Protoc Pharmacol* **73**, 14 39 11-14 39 20, doi:10.1002/cpph.2 (2016).
- 25 Webb, E. R. *et al.* Immune characterization of pre-clinical murine models of neuroblastoma. *Scientific reports* **10**, 16695, doi:10.1038/s41598-020-73695-9 (2020).
- 26 Digregorio, M. *et al.* The expression of B7-H3 isoforms in newly diagnosed glioblastoma and recurrence and their functional role. *Acta Neuropathol Commun* **9**, 59, doi:10.1186/s40478-021-01167-w (2021).
- 27 Wang, R. & Green, D. R. Metabolic checkpoints in activated T cells. *Nat Immunol* **13**, 907-915, doi:10.1038/ni.2386 (2012).

REVIEWER COMMENTS

Reviewer #2 (Remarks to the Author):

The authors have adequately responded to reviewer feedback and the revised manuscript is suitable for publication.

Reviewer #3 (Remarks to the Author):

I would like to thank the authors for addressing all my concerns in the revised manuscript. However, one of their responses raises concerns.

I had asked

'Figure 4: it is unclear if different donors were used for the functional analysis of CAR T cells' and the authors replied

'We used the same donor for the whole manuscript and clarified it in the figure legends'

This is very unusual and not standard - I highly recommend that at least key in vitro studies are performed with at least 3 healthy donors; otherwise, there are concerns regarding scientific rigor and reproducibility.

COMMENTS FROM THE REVIEWERS:

Reviewer #2 (Remarks to the Author):

The authors have adequately responded to reviewer feedback and the revised manuscript is suitable for publication.

- We appreciate enthusiastic comments from the reviewer.

Reviewer #3 (Remarks to the Author):

I would like to thank the authors for addressing all my concerns in the revised manuscript. However, one of there responses raises concerns.

I had asked 'Figure 4: it is unclear if different donors were used for the functional analysis of CAR T cells' and the authors replied 'We used the same donor for the whole manuscript and clarified it in the figure legends' This very unusual and not standard - I highly recommend that at least key *in vitro* studies are performed with at least 3 healthy donors; otherwise, there are concerns regarding scientific rigor and reproducibility.

- We thank the reviewer's enthusiastic comments and suggestions.
- We produced B7-H3 V_HH-based CAR-T cells derived from three individual healthy donors and examined their cytotoxicity *in vitro* on four different tumor cell lines (BxPC-3, Panc-1, IMR5, and NBEB) that were used for establishing tumor xenografts in mice. Our findings consistently demonstrate a markedly higher level of cytotoxicity exhibited by B7-H3 nanobody CAR-T cells compared to the control group (CD19 CAR-T) when co-incubated with tumor cell lines. We added the data in Supplementary Fig. 4C.

REVIEWERS' COMMENTS

Reviewer #3 (Remarks to the Author):

Thank you for providing the additional data. I would like to congratulate the authors to their excellent study.

REVIEWERS' COMMENTS

Reviewer #3 (Remarks to the Author):

Thank you for providing the additional data. I would like to congratulate the authors to their excellent study.

- We appreciate enthusiastic comments from the reviewer.